# Plagued by a cryptic clock: insight and issues from the global phylogeny of *Yersinia pestis*

Katherine Eaton[1,2], Leo Featherstone[3], Sebastian Duchene [3], Ann G. Carmichael [4], Nükhet Varlık[5],
G. Brian Golding[6], Edward C. Holmes [7] & Hendrik N. Poinar [1,2,8,9,10 ✉]

Plague has an enigmatic history as a zoonotic pathogen. This infectious disease will unexpectedly appear in human populations and disappear just as suddenly. As a result, a longstanding line of inquiry has been to estimate when and where plague appeared in the past. However, there have been significant disparities between phylogenetic studies of the causative bacterium, *Yersinia pestis*, regarding the timing and geographic origins of its reemergence. Here, we curate and contextualize an updated phylogeny of *Y. pestis* using 601 genome sequences sampled globally. Through a detailed Bayesian evaluation of temporal signal in subsets of these data we demonstrate that a *Y. pestis*-wide molecular clock is unstable. To resolve this, we developed a new approach in which each *Y. pestis* population was assessed independently, enabling us to recover substantial temporal signal in five populations, including the ancient pandemic lineages which we now estimate may have emerged decades, or even centuries, before a pandemic was historically documented from European sources. Despite this methodological advancement, we only obtain robust divergence dates from populations sampled over a period of at least 90 years, indicating that genetic evidence alone is insufficient for accurately reconstructing the timing and spread of short-term plague epidemics.

[1] McMaster Ancient DNA Centre, McMaster University, Hamilton, ON, Canada. [2] Department of Anthropology, McMaster University, Hamilton, ON, Canada. [3] The Peter Doherty Institute for Infection and Immunity, University of Melbourne, Melbourne, VIC, Australia. [4] Department of History, Indiana University Bloomington, Bloomington, IN, USA. [5] Department of History, Rutgers University-Newark, Newark, NJ, USA. [6] Department of Biology, McMaster University, Hamilton, ON, Canada. [7] Sydney Institute for Infectious Diseases, School of Medical Sciences, University of Sydney, Sydney, NSW, Australia. [8] Department of Biochemistry, McMaster University, Hamilton, ON, Canada. [9] Michael G. DeGroote Institute of Infectious Disease Research, McMaster University, Hamilton, ON, Canada. [10] Humans and the Microbiome Program, Canadian Institute for Advanced Research, Toronto, ON, Canada. ✉email: poinarh@mcmaster.ca

Plague has an impressively long and expansive history as a zoonosis of rodents. The earliest "fossil" evidence of the plague bacterium, *Yersinia pestis*, stems from ancient DNA studies which date its first emergence in humans to the Late Neolithic Bronze Age (LNBA) approximately 5000 years ago[1]. During this time, *Y. pestis* has dispersed globally on multiple occasions due to an ability to infect a variety of mammalian hosts[2] and ever-expanding trade networks[3]. Few regions of the ancient and modern world remain untouched by this disease, as plague has demonstrated a persistent presence on every continent except Australia and Antarctica[4]. There are three historically documented pandemics of plague: the First Pandemic (6th to 8th century CE)[5], the Second Pandemic (14th to 19th century CE)[6], and the Third Pandemic (19th to 20th century CE)[7]. The advent of each has been marked by a series of outbreaks, such as the medieval Black Death (1346–1353 CE), which is estimated to have killed more than half of Europe's population[8].

One long-standing line of inquiry in plague's evolutionary history has been estimating the timing, origins, and spread of these past pandemics. The most intensively researched events have been: (1) the first appearance of *Y. pestis* in human populations[9], (2) the onset and progression of the three pandemics[5,10,11] and (3) the inter-pandemic or "quiescent" periods where *Y. pestis* recedes into wild rodent reservoirs and disappears from the historical record[12,13,14]. Our knowledge of these events has deepened considerably in recent years, in part owing to technological advancements in the retrieval and sequencing of ancient DNA alongside new molecular clock dating methods[15].

Despite the intensive interest and methodological advancement, the rate and time scale of evolution in *Y. pestis* remains notoriously difficult to estimate. This is largely attributed to the substantial variation in evolutionary rates documented across the phylogeny[11,14]. As a result, considerable debate has emerged over whether *Y. pestis* has no temporal signal, thereby preventing meaningful rate estimates, or if some *Y. pestis* populations have such distinct rates that a species-wide signal is obscured[5,11,14,15]. This uncertainty has resulted in radically different evolutionary rate and date estimates between studies, with node dates shifting by several millennia and narrowing considerably[11,14] as additional ancient *Y. pestis* genomes are sampled as calibration points.

The geographic origins and progression of past pandemics are similarly contentious, particularly concerning mechanisms of spread and their underlying ecology. This contention concerns competing hypotheses about the relative importance of localized persistence versus long-distance reintroduction[16–18]. Among both sides of this issue, there is an expectation[18] that genomic evidence will play a significant role, and even optimism[19] that further sequencing of ancient strains might resolve the debate. However, no study to date has statistically evaluated whether *Y. pestis* genomes have sufficient geographic signal to confidently infer ancestral locations and spread.

To address these debates and obstacles, we: (1) curated and contextualized the most recent *Y. pestis* genomic evidence, (2) reviewed and critiqued our current understanding of plague's population structure, (3) devised a new approach for recovering temporal signals in *Y. pestis*, and (4) critically assessed the reliability of phylogeographic analyses. We ground our results and their interpretation using informative historical case studies to demonstrate the methodological and interpretive consequences. We anticipate these results will impact both retrospective and prospective studies of plague, which seek to date the emergence and spread of past pandemics as well as monitor the progression of ongoing outbreaks.

## Results and discussion
**Population structure**. To determine the population structure of *Y. pestis*, we first estimated a maximum likelihood phylogeny using 601 global isolates including 540 modern (89.9%) and 61 ancient (10.1%) strains (Methods). We rooted the tree using two genomes of the outgroup taxa *Yersinia pseudotuberculosis*. The alignment consisted of 10,249 variant positions exclusive to *Y. pestis*, with 3844 sites shared by at least two strains. Following phylogenetic estimation, we pruned the outgroup taxa from the tree to more closely examine the genetic diversity of *Y. pestis*. In Fig. 1A, we contextualize the global phylogeny using three nomenclature systems: the metabolic biovars, major branches, and historical time periods. In the following section, we compare and critique each system, identify any incongruent divisions and uncertainty, and explore an integrative approach for molecular clock analysis.

**Biovars**. The oldest classification system of *Y. pestis* is the biovar nomenclature that uses metabolic differences to define population structure. Accordingly, *Y. pestis* can be categorized into four classical biovars: *Antiqua* (ANT), *Medievalis* (MED), *Orientalis* (ORI), and *microtus/pestoides* (PE)[20,21]. Non-classical biovars have also been introduced, such as the *Intermedium* biovar (IN), which reflects a transitional state from *Antiqua* to *Orientalis*[22]. The biovar system is simple in application, as it largely focuses on two traits: the ability to ferment glycerol and reduce nitrate[21]. However, this simplicity is offset by the growing recognition of regional inconsistencies in metabolic profiles[23]. This is further exacerbated by the sequencing of non-viable, "extinct" *Y. pestis* strains for which metabolic sub-typing is challenging[5,9]. Researchers have responded to this uncertainty in a variety of ways, by extrapolating existing biovars[5] and creating new pseudo-biovars (PRE)[9]. Others have foregone the biovar nomenclature altogether in favor of locally developed taxonomies[23]. Despite extensive research, it remains unclear which metabolic traits, if any, can be used to classify *Y. pestis* into distinct populations on a global scale.

**Major branches**. In contrast to the biovar nomenclature which emphasizes phenotype, the major branch nomenclature focuses on genotype. This system divides the global phylogeny of *Y. pestis* into populations according to their relative position to a multi-furcation called the "Big Bang" polytomy[11]. All lineages that diverged prior to this polytomy are grouped into Branch 0 and those diverging after form Branches 1 through 4. Because this multifurcation plays such a central role in this system, there is great interest in estimating its timing and geographic origins[13,24].

**Time period**. Ancient *Y. pestis* genomes now represent a substantial portion of the known genetic diversity yet cannot be easily classified via direct metabolic testing. An alternative strategy has been employed that incorporates contextual evidence such as the sampling age, historical time period, and potential pandemic associations. In ancient DNA studies, the genetic diversity of *Y. pestis* is commonly divided into four time periods: the LBNA[9], the First Pandemic[5], the Second Pandemic[14], and the Third Pandemic[11] (Fig. 1B).

The key strengths of the time period nomenclature are twofold. First, it provides a foundation for interdisciplinary discourse, in which the genetic diversity can be contextualized and explained using relevant historical records. Second, this system effectively categorizes the historical outbreaks of plague recorded in Europe. This can be seen in Fig. 1, where the Bronze Age strains (0.PRE) in Europe are replaced by those of the First Pandemic (0.ANT4), which in turn are replaced by strains of the Second Pandemic (1.PRE). However, this "strength" comes at a cost, as this system is far less effective in describing plague populations outside of Europe and incurs two significant risks.

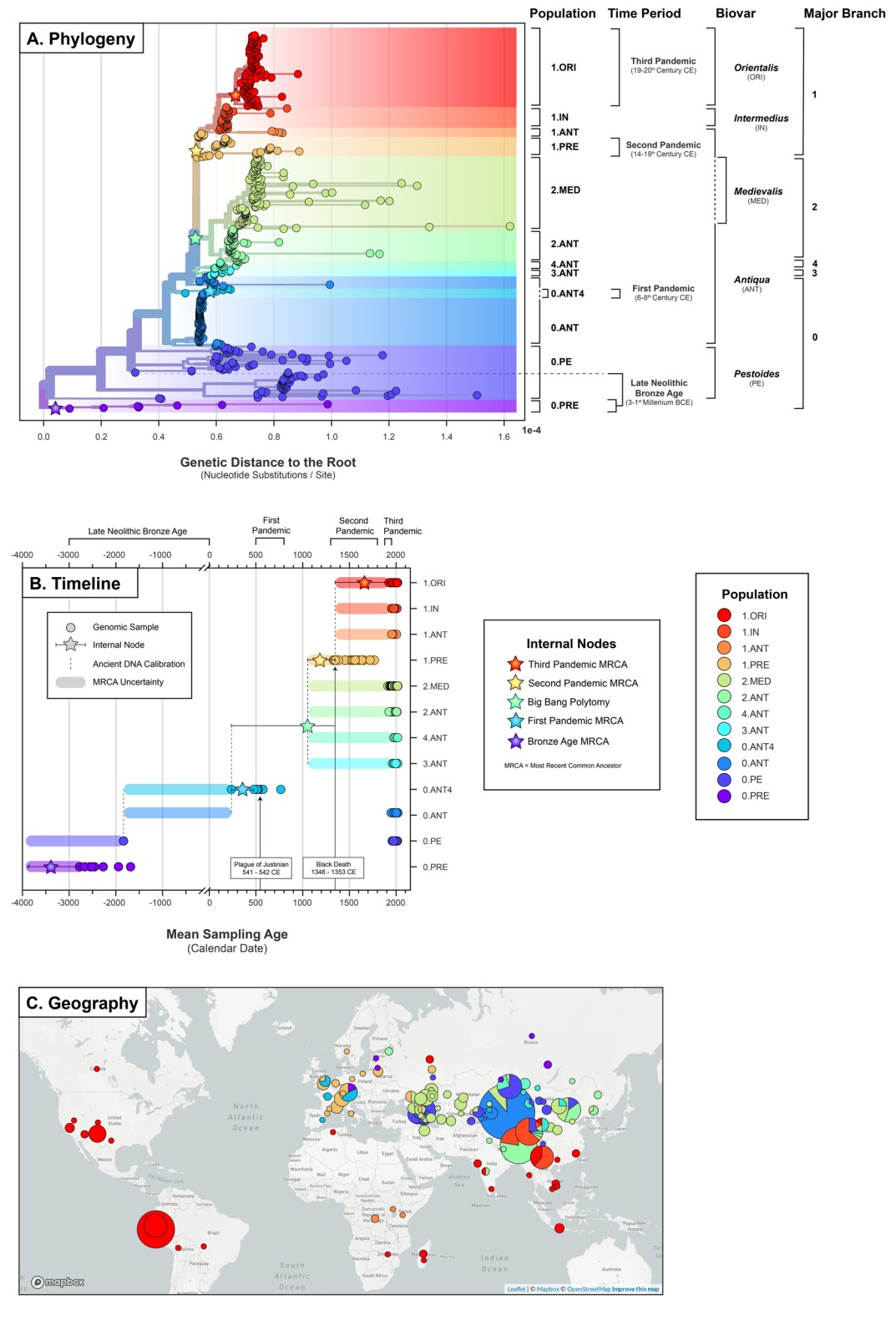

**Fig. 1 Global phylogeny of *Yersinia pestis*.** The phylogenetic and spatiotemporal diversity of 601 *Y. pestis* genomes. Populations were defined by integrating three nomenclature systems: the major branches, biovars, and time periods. **A** The maximum likelihood phylogeny of *Y. pestis* with strict bifurcations (no polytomies) and branch lengths scaled by genetic distance from the root in the number of nucleotide substitutions per site. The tree was rooted using two genomes of the outgroup taxa *Y. pseudotuberculosis*, which were pruned before visualization. **B** The mean sampling age of each genome with internal node dates bounded by ancient DNA calibrations. **C** The sampling location of each genome with coordinates standardized to the centroid of the associated province/state. The map image is copyright Mapbox and Open Street Maps used with permission. All images are produced by nextstrain.org are by CC BY.

The first risk is artificially grouping unrelated populations. Contemporaneous strains can have distinct evolutionary histories[25] even when originating from the same plague foci. The *Pestoides* (0.PE) and *Medievalis* (2.MED) biovars are informative examples, as these populations have co-existed in the Caucasus mountains since at least the 20th century (Fig. 1C). The second risk is artificially separating related populations. The Second and Third Pandemics were previously seen as mutually exclusive events dated to the 14th to 18th century, and the late 19th to mid-20th century, respectively[26]. Recent historical scholarship has contested this claim and demonstrated that these dating constraints are a product of a Eurocentric view of plague[6]. The Second Pandemic is now recognized to have extended into at least the 19th century[27,28] and the Third Pandemic is hypothesized to have begun as early as the 18th century[7,29]. Phylogenetic analysis reveals a degree of genetic continuity between these two events, as the Third Pandemic (1.ORI) is a direct descendant of 14th century strains from the Second Pandemic (1.PRE)[30]. However, uniquely derived lineages were also appearing at this time, as Second Pandemic strains sampled in 15–18th century Europe are phylogenetically distinct from those of the Third Pandemic. What remains unknown is the extent of temporal overlap, and as such, it is unclear where to draw the distinction between these pandemics using genetic evidence.

A final limitation is that several populations are curiously excluded from the pandemic nomenclature altogether. For example, Branch 2 populations emerged at the same time as, but separate from, the Second Pandemic and have been associated with high mortality epidemics[31]. In particular, the *Medievalis* population (2.MED) has dispersed throughout Asia (Fig. 1) with the fastest spread velocity of any *Y. pestis* lineage[7]. Despite its epidemiological significance, Branch 2 populations across Asia continue to be overlooked in the pandemic taxonomy of *Y. pestis*. As ancient DNA sampling strategies expand in geographic scope, and as more non-European historical sources are brought to bear, it will be important to consider how best to refashion the historical period nomenclature to encompass this diversity.

**An integrative approach**. There exists no current classification system which comprehensively represents the global population structure of *Y. pestis*. Instead, integrative approaches have been previously used in large comparative studies of *Y. pestis*[11,32]. We, therefore, take the intersection of the three taxonomic systems discussed previously, to define 12 populations where each constitutes a unique combination of a major branch, biovar, and time period (Fig. 1, Table S1). In the following sections, we highlight the novel insight and issues that arise when this population structure is explicitly incorporated into molecular clock models and phylogeographic reconstructions.

**Estimating rates of evolutionary change**. The extent of rate variation presented in our updated genomic data set is notably larger than those depicted in previous studies[14,32,33]. A root-to-tip regression of genetic distance as a function of sampling time reproduces the finding that substitution rates in *Y. pestis* are poorly represented by a simple linear model or "strict molecular clock" (Fig. 2A, B and Supplementary Fig. S1). We found a very low coefficient of determination ($R^2 = 0.09$) that indicates a large degree of unaccounted variation. This finding suggests that evolutionary change in *Y. pestis* may be more appropriately estimated using a "relaxed molecular clock", where rate variation is explicitly modeled. To test this hypothesis, we performed a Bayesian Evaluation of Temporal Signal (BETS)[34]. In brief, this method tested four model configurations including: (1) a strict clock, (2) a relaxed clock, (3) the inclusion of sampling ages, and (4) no sampling ages. Configurations with and without sampling ages

explicitly test for the presence of temporal signal. A comparison of the model marginal likelihoods, or Bayes factors, was then used to assess the degree of temporal signal.

We were unable to fit a single clock to the updated global diversity of *Y. pestis* using BETS. The Markov chain Monte Carlo (MCMC) traces exhibited poor sampling of parameter space (effective sample size, ESS < 200) across all model configurations, even when we reduced sources of variation by removing tip date uncertainty, fixing the tree topology, and removing up to 70% of the genomes. The poor performance of a single clock model is consistent with several other studies, in which low ESS values were observed[9] and divergence dates could not be estimated[5]. A single clock model is not a viable approach for *Y. pestis*, as there is excessive rate variation across the global phylogeny, which likely explains node-dating disparities between previous studies[9,11,32].

In contrast to the single clock approach, we observed substantial improvements when each population was assessed independently. All model parameters in our Bayesian analysis demonstrated MCMC convergence with ESS values well above 200, and we detected temporal signal in 9 out of 12 *Y. pestis* populations (Table S2). Several of these appeared more clock-like than others, which was observed through the root-to-tip regression and Bayesian rate estimation. For example, we found rate variation to be low in the Bronze Age ($R^2 = 0.92$), moderate in the Second Pandemic ($R^2 = 0.76$) and high in *Medievalis* ($R^2 > = 0.02$). Our results indicate that population specific models are a more effective approach for estimating substitution rates across the global phylogeny.

To demonstrate the application of our molecular clock method and the interpretive consequences, we explored three outcomes as case studies. First, as a control, we examined *Y. pestis* populations that had (i) no temporal signal. These "negative" cases inform us about the minimum sampling time, also known as the phylodynamic threshold[35] required to obtain robust temporal estimates in *Y. pestis*. Second, we examined populations with (ii) irreproducible estimates between studies, such as the time to (the) most recent common ancestor (tMRCA). We discuss how sampling bias drives this outcome, and how it can be identified and corrected with ancient DNA calibrations where available. Finally, we identify the populations with the most (iii) informative rates and dates. We discuss how these molecular dates change our understanding of pandemic "origins" and complement recent historical scholarship.

**No temporal signal**. We found several *Y. pestis* populations with no detectable temporal signal that include the *Intermedium* (1.IN) and *Antiqua* biovars (2.ANT, 3.ANT). Despite being sampled over a period as long as 84 years (2.ANT), these populations have not accumulated sufficient evolutionary change to yield informative divergence dates. This limited diversity is identifiable in the maximum likelihood phylogeny as populations with the highest density of nodes sitting close to their corresponding root nodes (Supplementary Fig. S2). Out of caution, we also consider the rates and dates associated with the *Antiqua* population 4.ANT to be non-informative, as it has a similar node distribution, with a smaller number of samples ($N = 12$) collected over an even shorter time frame (38 years).

Our results show that for robust temporal estimates to be obtained, *Y. pestis* must be sampled over multiple decades at minimum, while minimizing population structure, as in the whole data set described above. This time frame is largely consistent with the finding that *Y. pestis* has one of the slowest substitution rates observed among bacterial pathogens[15]. Here, we found that all populations had a median rate of less than 1 substitution per year (Fig. 2C, S2, Table S4), with the lowest rate

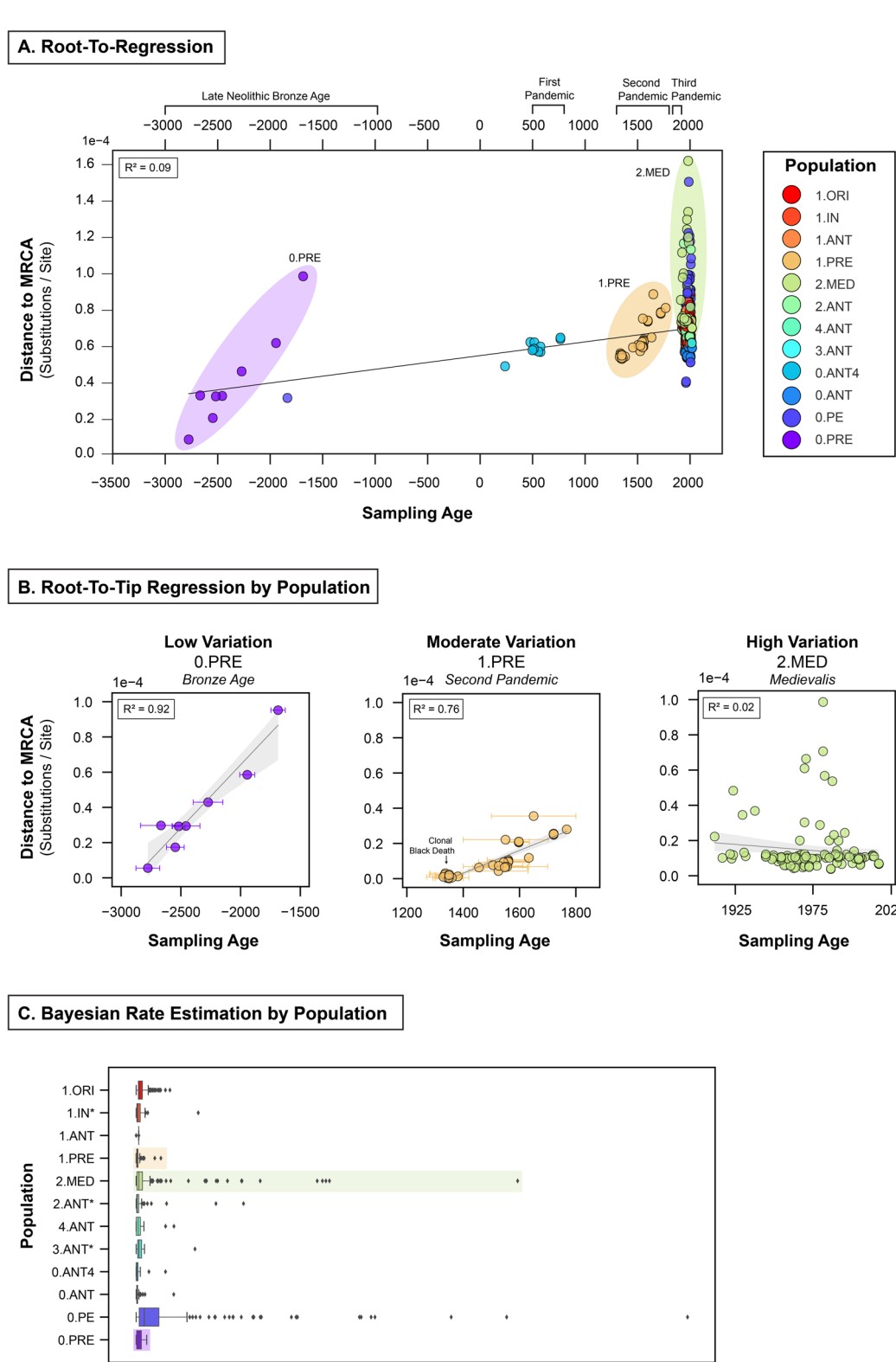

**Fig. 2 Substitution rate variation in *Y. pestis*. A** A root-to-tip regression on mean sampling age using all genomes from the maximum likelihood phylogeny. **B** A root-to-tip regression on mean sampling age by population. The distance to the population MRCA was calculated using subtrees extracted from the maximum likelihood phylogeny. **C** Bayesian substitution rates within and between populations. For each branch in the maximum clade credibility (MCC) trees, we extracted the estimated substitution rate (subs/site/year) and converted this to subs/year based on an alignment of 4,229,098 genomic sites.

in *Antiqua* (0.ANT) at 1 substitution every 14.1 years ($1.7 \times 10^{-8}$ subs/site/year) and the highest rate in *Pestoides* (0.PE) at 1 substitution every 1.1 years ($2.1 \times 10^{-7}$ subs/site/year). In application, this means that *Y. pestis* lineages often cannot be differentiated until at least several decades have passed, a concept referred to in the literature as the phylodynamic threshold[34].

The phylodynamic threshold has been rigorously explored in other pathogens, such as SARS-CoV-2[36], but not explicitly in *Y. pestis*. The challenges in reconstructing intra-epidemic plague diversity have been noted previously. For example, several isolates from the Second Pandemic dated to the medieval Black Death (1346–1353) are indistinguishable clones[30], extinguishing any hope of reconstructing its spread from genetic evidence alone. Our median rate estimation for the Second Pandemic (1.PRE) of 1 substitution every 9.5 years ($2.5 \times 10^{-8}$ subs/site/year) is congruent with this finding. The clonal nature of the Black Death is not an exceptional event, but rather the norm based on the sampling time frame. Our results highlight a significant limitation and cautionary note for plague research, as genetic evidence alone is not suitable for reconstructing the timing of short-term, episodic epidemics.

**Irreproducible estimates.** We observed two populations with detectable temporal signal associated with substantial node-dating conflicts: the *Pestoides* (0.PE) and *Antiqua* (0.ANT) biovars: both of which are paraphyletic. Conflicts were identified by comparing their estimated time to the most recent common ancestor (tMRCA) to that of their descendant populations. For example, the First Pandemic (0.ANT4) is a descendant clade of the larger *Antiqua* (0.ANT) population based on the maximum likelihood phylogeny (Fig. 3). We would expect the tMRCA of the ancestral 0.ANT to pre-date the First Pandemic, for which ancient DNA calibrations are available. However, the tMRCA of 0.ANT is far too young (95% HPD: 1357–1797 CE), and incorrectly post-dates the tMRCA of 0.ANT4 (95% HPD: 39–234 CE) by more than a millennium. This outcome is somewhat paradoxical, as these populations have robust temporal signal and yet a critical examination of their divergence dates reveals they are unreliable.

This conflicting pattern has been previously described and attributed to sampling bias[37,38], specifically, insufficient sampling of basal branches and the presence of extensive rate variation. The two affected populations, *Antiqua* (0.ANT) and *Pestoides* (0.PE), have a low density of nodes at their roots in the maximum likelihood phylogeny (Supplementary Fig. S2). This pattern is also observed in another *Antiqua* population (1.ANT) which has a

small sample size ($N = 4$) and has previously been linked to rate acceleration events[11]. The dates associated with these three populations (0.PE, 0.ANT, 1.ANT) should be considered non-informative.

These node-dating issues reveal a clear limitation in our approach of estimating divergence times from population-specific models. Defining *Y. pestis* population by time periods has adverse effects, as ancient plague genomes can serve as crucial calibration points for rate changes that are otherwise unsampled in extant populations. In populations with poorly sampled basal branches (0.PE, 0.ANT, 1.ANT) we expect an optimization approach to be better suited, in which a few closely related populations are merged or select ancient DNA calibrations are introduced[25]. Otherwise, divergence dates in these populations tend to be overly young, sometimes by more than a 1000 years[33], and are difficult to replicate between studies (Table 1).

The inability to infer divergence dates due to sampling bias also has several historical implications. Perhaps the most significant concerns the emergence of plague in Africa which makes up 90% of all modern plague cases[39], yet for which there remains not a single ancient sequence. Little progress has been made in sampling extant African plague diversity, with this region represented by only 1.5% (9/601) of all genomes. Furthermore, the oldest genetic evidence of African plague comes from the 1.ANT population, which has only four representative strains. Despite this sparse sampling, researchers have repeatedly attempted to use genomic evidence to date the first appearance of *Y. pestis* in Africa[11,32,33]. The result is a complete lack of congruent dates for this event, as the majority of tMRCA estimates for 1.ANT do not overlap (Table 1). These divergence dates are of limited value for historical interpretation[33,40,41] and should be treated with great skepticism.

**Informative rates and dates.** Excluding populations with no detectable signal, we identified five populations with potentially informative rates and dates. These include the Bronze Age (0.PRE), *Medievalis* (2.MED), the First Pandemic (0.ANT4), the Second Pandemic (1.PRE), and the Third Pandemic (1.ORI). The Bronze Age marks the first identified appearance of *Y. pestis* in humans, and the three pandemics, along with *Medievalis*, are historically associated with high mortality and rapid spread[7]. Due to this epidemiological significance, these five populations have been sampled over the longest time frames, ranging from 92 years for the Third Pandemic (1.ORI) to 1250 years for the Bronze Age (1.PRE). This affirms the importance of long-term heterochronous sampling for *Y. pestis*, made possible through the retrieval of ancient DNA[10] and recent sequencing of early 20th century culture collections[31]. By curating and contextualizing this new heterochronous data, we were able to detect temporal signal in extant *Y. pestis* populations without the use of ancient DNA calibrations for the first time.

Our estimates of the tMRCA for the First and Second Pandemics share a common theme, in that the genetic origins potentially pre-date the appearance of plague in traditional (i.e., European) historical narratives (Table S5). For example, the earliest textual evidence of the Second Pandemic (1.PRE) in Europe comes from the Black Death (1346)[8]. However, we estimate the mean tMRCA of this population to be earlier, between 1214 and 1315 CE. Similarly, the first recorded outbreaks of plague during the First Pandemic (0.ANT4) come from the Plague of Justinian (541 CE)[42]. Instead, we estimate that the strains of *Y. pestis* associated with this pandemic shared a common ancestor between 272 and 465 CE.

One explanation for these disparate timelines is uncertainty in estimating the root position when only samples from within a

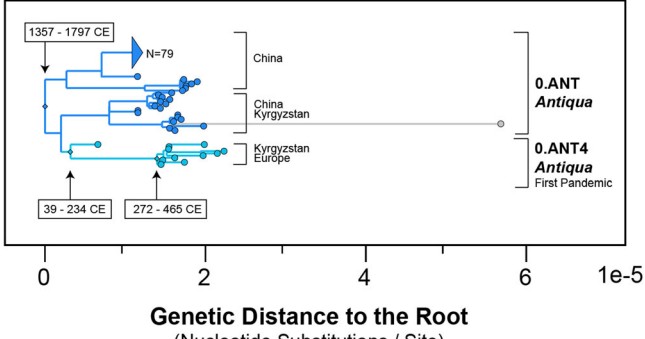

**Fig. 3 Ancestor-descendant relationships in the maximum likelihood phylogeny reveal tMRCA conflicts between Antiqua (0.ANT) and the First Pandemic (0.ANT4).** Node dates (95% HPD) were estimated from the Bayesian analysis, where each population was assessed independently. Grey branches indicate outliers, as defined by the 90% confidence interval of external branch lengths from all populations.

**Table 1 Bayesian estimates of the time to most recent common ancestor (tMRCA) across *Y. pestis* studies.**

| Category | Population | Morelli et al. 2010 | Cui et al. 2013 | Rasmussen et al. (2015) | Pisarenko et al. 2021 | Spyrou et al. 2019 | This Study |
|---|---|---|---|---|---|---|---|
| Informative Dates | 1.ORI | −326, 1793 | 1735, 1863 | – | 1744, 1842 | – | 1806, 1901 |
| No Temporal Signal | 1.IN | −2388, 1606 | 1500, 1750* | – | 1791, 1897 | – | 1651, 1913 |
| Irreproducible Estimates | 1.ANT | −4909, 1322 | 1377, 1650 | 1045, 1480 | 1483, 1704 | – | 1655, 1835 |
| Informative Dates | 1.PRE | – | 1312, 1353 | 901,1343 | – | 1182, 1333 | 1214, 1315 |
| Informative Dates | 2.MED | −583, 1770 | 1550, 1800* | – | 1413, 1653 | – | 1560, 1845 |
| No Temporal Signal | 2.ANT | −3994, 1460 | 1550, 1800* | – | 1373, 1628 | – | 1509, 1852 |
| No Temporal Signal | 4.ANT | – | 1200, 1700* | – | 1611, 1816 | – | 1848, 1968 |
| No Temporal Signal | 3.ANT | – | 1450, 1850* | – | 1531, 1742 | – | 1769, 1947 |
| Irreproducible Estimates | 0.ANT | −6857, 1199 | 100, 1100* | – | 1033, 1435 | – | 1357, 1797 |
| Informative Dates | 0.ANT4 | – | – | −400,315 | – | – | 39, 234 |
| Irreproducible Estimates | 0.PE | −26641, −598 | −4394, 510 | – | −377, 499 | – | 1573, 1876 |
| Informative Dates | 0.PRE | – | – | −5007, −3006 | – | – | −3098, −2786 |

Uncertainty surrounding the tMRCA is represented by the 95% highest posterior density (HPD) interval. A dash indicates the study did not incorporate genomes from the population. No Temporal Signal indicates a population had no detectable temporal signal according to a BETS test. The Irreproducible Dates category defines a population with node-dating conflicts attributed to sampling bias, despite having detectable temporal signal. Informative Dates reflects a population with detectable temporal signal and no node-dating conflicts.
*Visually estimated from the published time-scaled phylogeny.

population are used. This pertains specifically to the Second Pandemic (1.PRE), where basal sequences have near-zero branch lengths in the maximum-likelihood phylogeny (Fig. 1A) which leads to poor branch support in the maximum-clade-credibility tree (Supplementary Fig. S3). The most basal strain (LAI009) in the maximum-clade credibility tree is placed in a more derived position (farther from the root). This is surprising given that it is only a single SNP different (at the core genome level) from the Black Death clonal strains that emerge in 1347/8. However, we conducted analyses in which the tree topology of this clade was fixed and LAI009 specified as an outgroup: these produced nearly identical posterior densities for the age of the root.

A second explanation for our earlier dates is tip date uncertainty. The radiocarbon estimates for the majority of ancient *Y. pestis* samples have confidence intervals of ±50 years or more. As we only used the mean sampling age for molecular clock models, it is possible that the true tMRCA intervals are larger and do overlap with historical estimates. How much uncertainty can be included in molecular clock models for *Y. pestis*, while still achieving convergence of parameter estimates, remains to be tested.

A third explanation is geographic sampling bias, as western European sources dominate both the genetic and historical record. Recent historical scholarship has contested Eurocentric timelines[28,43] by demonstrating the presence of plague in western Asia far earlier than previously thought. Arabic historical chronicles suggest that the Second Pandemic may have begun as early as the 13th century[44]. Genetic dating appears to support these historical critiques, by expanding the timelines of past pandemics to make space for more diverse historical narratives.

Recently published aDNA data[45] characterizes *Y. pestis* genomes from two individuals from cemeteries located near Lake Issyk-Kul, Kyrgyzstan. These genomes fall on the node of the polytomy and are putatively dated to 1338–1339 and help constrain the clock by yielding an MRCA for branches 1–4 to between 1316–1345, which postdates our estimated age range of 1214–1315 for just the 1.PRE— Second Pandemic clade. Clearly, at this proximity to the true node, the variance on the clock has a huge effect on estimation of a MRCA. The only way to properly calibrate it is with well dated sequences, such as those from skeletal remains at Lake Issyk-Kul[45].

In contrast to the ancient pandemics, our temporal estimates of the Third Pandemic were more closely aligned to the historical

evidence. We estimated that isolates from the Third Pandemic (1.ORI) shared a common ancestor between 1806 and 1901 CE, which aligns well with the timeline as reconstructed from epidemiological reports. Highly localized plague cases began appearing in southern China and (1772–1880) and later spread globally out of Hong Kong (1894–1901)[7,46,47]. Our estimate also overlaps with the majority of previous studies, although it is the youngest tMRCA to date (Table 1, Supplementary Table S5). This comparison demonstrates the reproducibility of our estimate but also reveals how the "origin" story of the Third Pandemic continues to change. The phylogenetic root once estimated to be as old as 326 BCE[32] is now resolved to be much younger (19th century CE). This younger date is particularly intriguing, as a major epidemiological transition occurred in the 19th century with the reemergence of several other notable pathogens[48]. Reconstructing the evolutionary history of the Third Plague Pandemic may not only inform us about the epidemiology of plague, but contribute to a broader understanding of the factors that led to reemerging diseases in the modern era[49,50].

Even less is known about the *Medievalis* population whose strains were hypothesized to be responsible for plague outbreaks in the Caspian Sea region which reoccurred throughout the 19th and 20th centuries[31]. We estimated the tMRCA of *Medievalis* (2.MED) to be between 1560 and 1845 CE, which overlaps with all previously published estimates (Table 1, Supplementary Table S5). While this population was once thought to have emerged as early as 583 BCE, there is now growing consensus that the earliest possible emergence was in the 16th century CE. Interestingly, the Caspian Sea region appears to be a nexus of plague as the only known area where the distributions of both European and Asian *Y. pestis* strains overlap (Fig. 4). This raises the interesting possibility that distinct populations of *Y. pestis* were co-circulating during the Second Pandemic, a hypothesis that ancient DNA from *Medievalis* could help elucidate. In the absence of direct genetic sampling, an alternative approach is to infer the ancestral locations and spread of plague using phylogeographic analysis.

**Estimating ancestral locations and spread.** Phylogeographic inference relies heavily on the degree of geographic signal, which can be thought of as the extent to which phylgenetic relationships

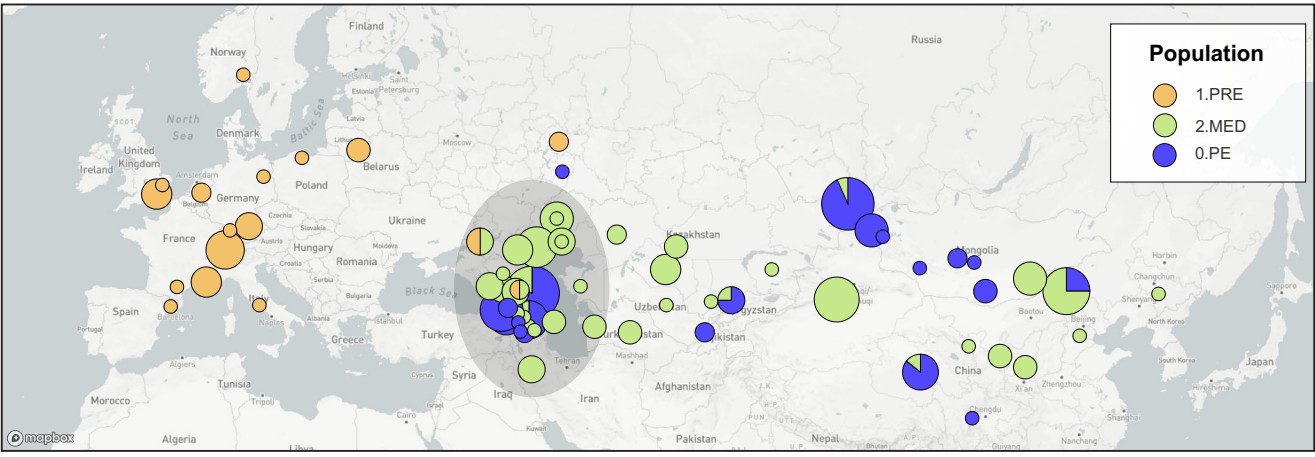

**Fig. 4 The geographic distributions of the Second Pandemic; (1.PRE), Medievalis (2.MED), and Pestoides (0.PE) populations.** The sampling location of each genome was standardized to the centroid of the associated province and/or state. The map image is copyright Mapbox and Open Street Maps used with permission. All images are produced by nextstrain.org are by CC BY.

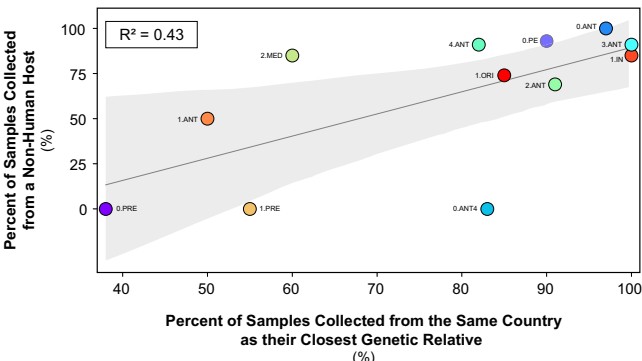

**Fig. 5 Association between host and country of origin.** Linear regression of host associations. A linear regression of host-associations on the degree of geographic structure at the country level.

correlate with sampling locations. To assess this in *Y. pestis*, we tested whether phylogenetic relationships correlate with sampling locations. We identified the closest genetic relative of every genome in our data set, using the maximum likelihood phylogeny. We then recorded whether these genomes were collected from the same location at three levels of resolution: (1) continent, (2) country, and (3) province. As a statistical measure of geographic structure, we reported the percent of genomes that had a closest relative sampled from the same location (Table S6).

The majority of *Y. pestis* populations (6/12) were localized to a single continent and their samples were often collected from the same location as their closest genetic relative (Supplementary Fig. S4). Of those distributed across multiple continents, geographic structure ranged from 76 to 99%. At the country level, the degree of geographic structure dropped drastically in some populations (Bronze Age 0.PRE: 38%) while remaining stable in others (3.ANT: 100%). The inverse of this pattern appeared at the province level, where *Antiqua* (3.ANT) dropped to 45% while the Bronze Age (0.PRE) was unchanged. As expected, geographic structure decreases at finer resolutions by the extent to which it varies by population.

The factors which appear to govern these patterns are wide-ranging, but primarily concern movement of the pathogen. One striking aspect is the difference in host composition between populations driving this signal. We observed a correlation ($R^2 = 0.43$) between the degree of geographic structure and the percentage of samples collected from a non-human host (Fig. 5).

Populations primarily sampled from rodents and arthropod vectors had far more geographic structure than those found exclusively in humans. This is epidemiologically consistent with the greater mobility of human populations, which disrupt geographic clustering via long-distance spread.

Another factor is the difference in substitution rates relative to migration rates. In populations that are spread faster (locations/year) than they evolve (substitutions/year), geographic structure decreases as identical isolates (clones) are found in different locations. Lineages of the Second Pandemic (1.PRE) exemplify this, as clonal isolates have been sampled across multiple countries[14], leading to uncertainty regarding the routes of spread. Along similar lines is the disparity in the sizes of locations across the globe used in the analyses. China, for example, is approximately the same size (0.94) as the European continent. Thus, at the country level, plague populations in Asia are sampled over fewer locations and have stronger geographic structure. An informative comparison is *Antiqua* (3.ANT) with 100% structure across China and Mongolia, and the Second Pandemic (1.PRE) with 55% structure across 11 European countries. Plague populations that are distributed over multiple, smaller locations have less geographic structure, leading to greater uncertainty when inferring past migrations.

These observations suggest that phylogeographic inference is best suited to populations that are slow-spreading and/or rapidly-evolving, a significant problem for *Y. pestis*, as it can be both a rapidly-spreading and slow-evolving pathogen[15]. To explore how this feature impacts our ability to infer ancestral locations for *Y. pestis* in the past, we independently fit three discrete migration models[51] to the maximum likelihood phylogeny using the sampling locations by: (1) continent, (2) country, and (3) province (Supplementary Table S7). We chose this approach, over one where time is coestimated (i.e., a molecular clock) because the ages of some samples have uncertainties associated with them and due to the large rate variation across the entire tree, which would be a large source of error in migration rates and events over time. For each internal node, we extracted the ancestral location with the highest likelihood given the data (Supplementary Table S7). To explore whether genomic evidence can provide meaningful geographic estimates, we compared two case studies: the Third Pandemic, which serves as a "control" for our phylogeographic analysis, and the Second Pandemic, where the origins and spread remain contentious due to limited non-European historical evidence. Importantly, we focus our interpretation of migration events to those that have high statistical

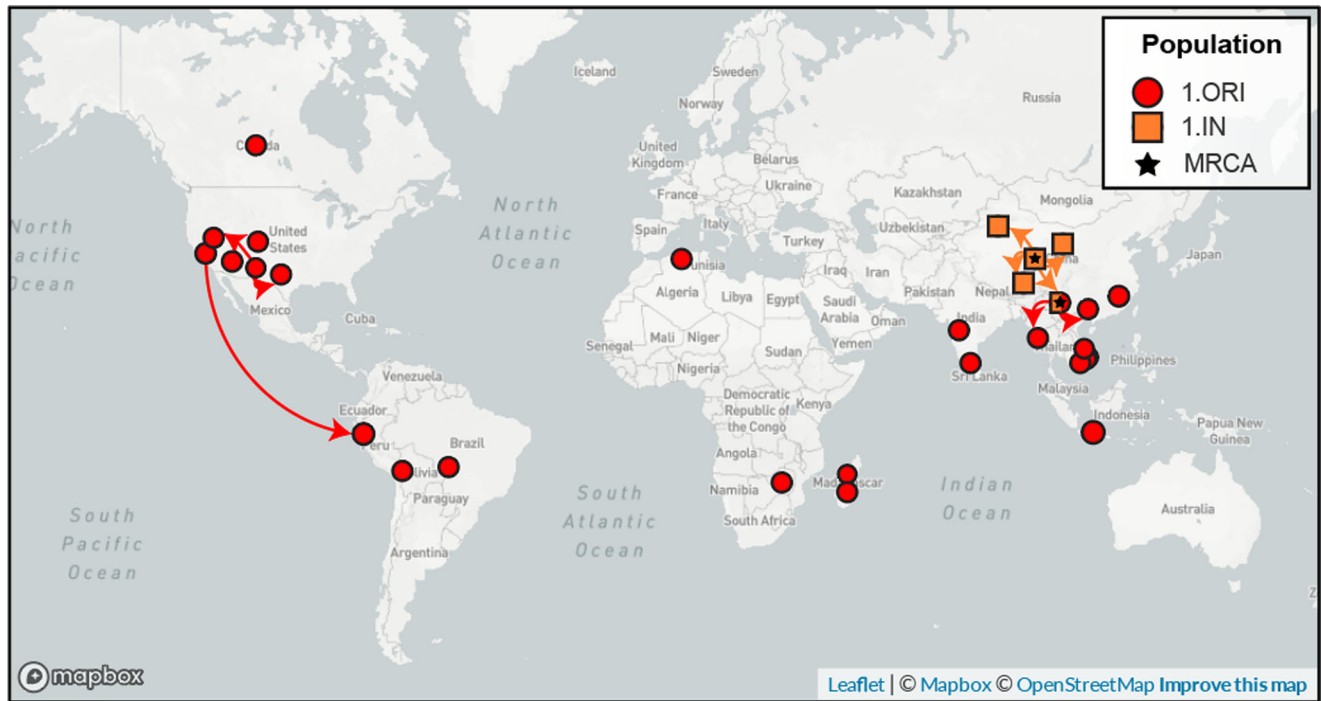

**Fig. 6 Geographic origins and spread of the Third Pandemic (1.ORI) and the intermedium (1.IN) population.** Ancestral locations were estimated by fitting a discrete migration model to the maximum likelihood phylogeny using sampling locations by province. Arrows reflect the directionality of spread, but not the precise route taken. High-confidence migrations estimated with an ancestral likelihood greater than or equal to 0.95 and a branch support bootstrap of greater than or equal to 95% are shown. copyright Mapbox and Open Street Maps used with permission. All images are produced by nextstrain.org are by CC BY.

support, with high bootstrap (topology inference) and discrete trait reconstruction (phylogeographic inference). In practice, low statistical support in these events means that the data are not sufficiently informative about migration pathways, and thus their interpretation can be misleading.

**The Third Pandemic (19–20th century CE).** The Third Pandemic of plague was closely monitored by contemporary researchers[52]. As a result, the geographic origins are well-documented and firmly established. Highly localized plague cases first appeared in Yunnan, China (1772–1800), later spreading throughout the province (1800–1880)[7,46]. Plague then traveled eastwards to the coast (1880–1900), where it dispersed globally out of Hong Kong (1894–1901)[53].

We estimate that the Third Pandemic (1.ORI) diverged from an ancestral *Intermedium* (1.IN) population located in Yunnan, China (confidence: 1.00) (Fig. 6, Supplementary Fig. S5, S6). Plague then rapidly diversified, after which new lineages appeared in North America (confidence: 0.99), South America (confidence: 1.00). Due to the unresolved branching structure, we could not confidently estimate the routes of this initial dispersal. The migrations that could be reconstructed all occurred post this radiation, and included endemic cycling in Southeast Asia (China, Myanmar) and in North America (USA).

The strength and specificity of our estimated origin is striking, given that we could not confidently locate the ancestral location nor divergence for any other population (Supplementary Fig. S7). This may be because the Third Pandemic (1.ORI) is a direct descendant of the *Intermedium* (1.IN) population, which has strong geographic structure at the province level (87%). In addition, isolates from Yunnan fall both basal to, and within, the known diversity of the Third Pandemic (1.ORI). This combination provides strong evidence of the geographic origin, which is congruent with the historical narrative. This level of precision was

only possible due to the extensive sampling of non-human hosts. Yunnan is solely represented by rodent and arthropod samples ($N = 18$) and therefore this reservoir would be entirely invisible if only human isolates were used. Like others[19], we caution that the presence and location of rodent reservoirs should not be inferred from phylogenetic evidence alone. Instead, new modeling approaches have been developed[54] that could leverage multi-disciplinary sources[47] to correct for sampling biases in the genomic data.

**The Second Pandemic (14–19th century CE).** In comparison to the Third Pandemic, there is far less surviving historical evidence from the Second Pandemic. Historians have identified early accounts of plague appearing in 1346 in the Golden Horde, which encompasses Central Asia and Eastern Europe[55]. The disease then appears to have spread southward through the Caucasus to reach Western Asia, and westward to the Crimea, from which it dispersed throughout Europe, the Middle East, and North Africa. Plague recurred for several centuries in these territories, with successive waves varying in scale from localized epidemics to continent-wide outbreaks[8]. In continental Western Europe, plague receded after 1720 and would not re-emerge again until 1899[56], while in Western Asia the disease never disappeared[28].

We estimated that the Second Pandemic (1.PRE) diverged from an ancestral population located (0.ANT) in China (probability: 0.93) as part of the stem leading up to the "Big Bang" polytomy (Fig. 7, Supplementary Fig. S4). The ancestral province was poorly resolved, with the most likely location being near Xinjiang (confidence: 0.64) which includes the Tien Shan mountains (Supplementary Fig. S5) which is in congruence with recently published data[45]. The location of the Second Pandemic MRCA was also uncertain and estimated to be in Russia (confidence: 0.63), specifically in Tatarstan (confidence: 0.37) (Supplementary Fig. S7) which was part of the Golden Horde. However, these low

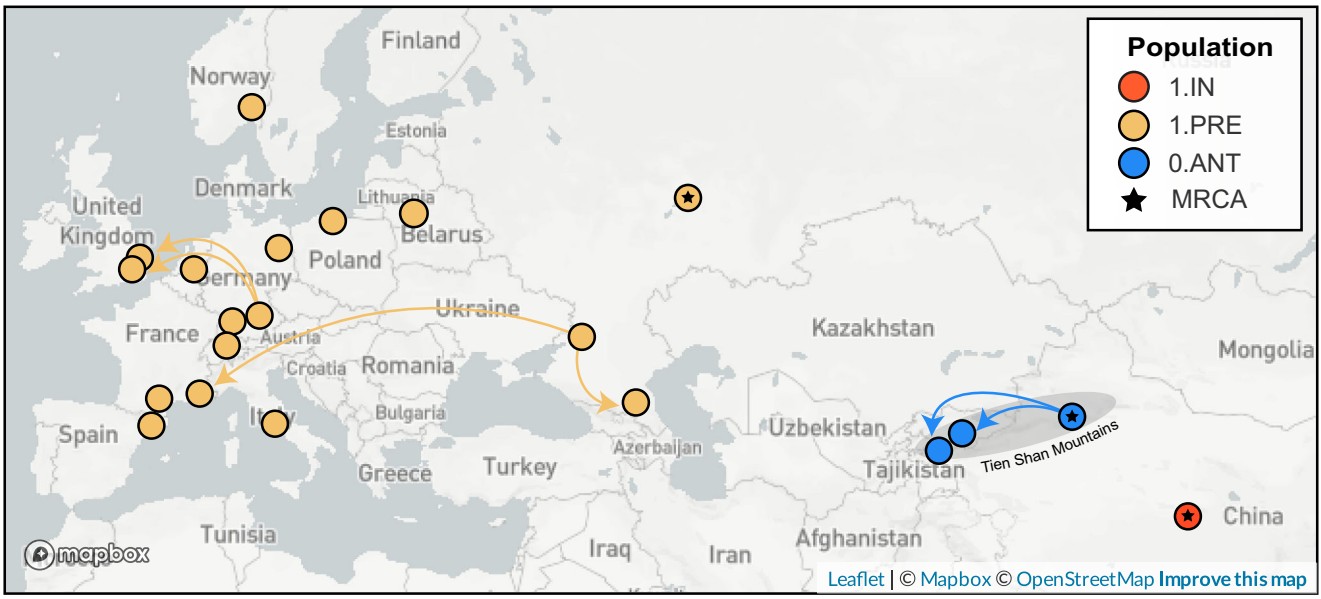

**Fig. 7 Geographic origins and spread of the Second Pandemic (1.PRE), the ancestral Antiqua (0.ANT) and descendant Intermedium (1.IN) populations.** Ancestral locations were estimated by fitting a discrete mugration model to the maximum likelihood phylogeny using sampling locations by province. Arrows reflect the directionality of spread, but not the precise route taken, for high-confidence migrations estimated with an ancestral likelihood greater than or equal to 0.95 and a branch support bootstrap of greater than or equal to 95%.

likelihoods indicate that our estimated origins are poorly supported by the data. With regards to spread only four migrations could be confidently inferred (confidence $p >= 0.95$) across the full sampling time frame (500 years). The available genetic data, therefore, provides little definitive evidence as to the spread of plague during the Second Pandemic.

This then begs the question of whether more ancient DNA samples will improve these geographic estimates. As it currently stands, the relationships between all countries could not be resolved during the 14th century, nor among the Baltic states sampled in the 15th century, or between England and Russia in the 17th century. Furthermore, the historical evidence indicates that plague often spread to multiple countries, if not continents, in the span of a decade[57]. This migration rate is far higher than the substitution rate of the Second Pandemic (1.PRE), which accumulates 1 mutation every 9.5 years. The genomic data alone does not have sufficient resolving power to reconstruct the spread of short-term, episodic waves of plague. Instead, this evidence is best used in conjunction with higher-resolution historical evidence[37,58].

**Conclusions**

We sought to contribute to five lines of debate concerning the evolutionary history of *Yersinia pestis*. The first, was whether *Y. pestis* has sufficient temporal signal (i.e., molecular clock) to accurately estimate rates and dates. Accordingly, we found that a species-wide clock model was methodologically unstable and did not lead to reproducible estimates. However, we observed significant improvements when each *Y. pestis* population was assessed independently. We, therefore, recommend this approach for future studies, as the full global diversity of *Y. pestis* can be utilized without down-sampling.

Second, we explored the minimum sampling time frame for *Y. pestis* that yields informative rates and dates. The lowest substitution rate was observed in *Antiqua* (0.ANT) with a median rate of 1 substitution every 14.1 years, meaning that some *Y. pestis* lineages cannot be differentiated until several decades have passed. In addition, we found no temporal signal in several

populations (1.IN, 2.ANT, 3.ANT) which have been sampled over a period as long as 84 years. Genetic evidence alone may not be suitable for reconstructing the timing of short-term, epidemic events of plague.

In the third instance we tackled node dating disparities between studies. We explored how phylogenetic sampling bias drives this outcome, and how it can be detected and remedied with ancient DNA calibrations. In particular, we focused on the non-overlapping tMRCA estimates of the first appearance of *Y. pestis* in Africa (1.ANT). Until sampling strategies diversify, we caution that the published divergence dates for this population, and several others (*Antiqua* 0.ANT, *Pestoides* 0.PE), are of limited value for historical interpretation.

The fourth point revolved around the timing of past pandemics. We observed a common theme in which the genetic dates (tMRCAs) of pandemic *Y. pestis* potentially pre-date the historical dates by several decades or centuries (Table 1 and S5). For example, we estimated the tMRCA of the Second Pandemic to be between 1214 and 1315 CE which pre-dates the Black Death (1346–1353 CE) and pre-dates a recent estimate of the MRCA for the polytomy from *Yersinia pestis* genomes reconstructed from Issyk-Kul, Kyrgyzstan[45]. Refinement of these early events and nodal MRCA's will require more ancient DNA data as the clock is too variable for this resolution. Similarly, we estimated the tMRCA of the First Pandemic to be between 272 and 465 CE, also pre-dating the Plague of Justinian (541 CE). We discussed this disparity in light of methodological concerns such as phylogenetic uncertainty in root positioning, radiocarbon dating uncertainty in tip dates, and geographic sampling biases that have historically favored European sources. We anticipate that additional phylogenetic analyses and samples from non-European locations will add greater clarity to how the timelines of past pandemics can be adjusted to include more diverse historical narratives.

Finally, we asked whether *Y. pestis* has sufficient geographic signal to accurately infer ancestral locations and spread. As expected, geographic structure diminished at finer resolutions but also varied by population. We found that the geographic origins of the Third Pandemic (1.ORI) were unambiguously inferred to be in Yunnan province, China (likelihood = 1.00) and attributed

this to comprehensive sampling of rodent reservoirs. In contrast, we demonstrated how the origins and spread of the Second Pandemic (1.PRE) cannot be resolved from genetic evidence alone, as this population is exclusively sampled from human remains which have high mobility. In isolation, *Y. pestis* genomic evidence may be unsuitable for inferring point migrations and the directionality of spread. Alternatively, new methods which incorporate non-genetic evidence, such as outbreak case-occurrence records[37], into phylogeographic analysis presents an exciting avenue for interdisciplinary collaboration, as explicitly integrative models will complement the strengths of genetic and historical evidence, while mitigating their respective weaknesses.

## Methods

**Data collection**. *Y. pestis* genome sequencing projects were retrieved from the National Center for Biotechnology Information (NCBI) databases on 2020 January 01 using NCBImeta (v0.7.0)[59]. 1657 projects were identified and comprised three genomic types. 1473 projects came from isolates sampled during the 20th and 21st centuries, which we label as "modern". Of these, (i) 586 projects were available as assembled genomic contigs (FASTA), and (ii) 887 were only available as unassembled sequences (FASTQ). An additional (iii) 184 projects came from skeletal remains with sampling ages older than the 19th century, which we label as "ancient". The 887 modern unassembled genomes were excluded from this project, as the wide variety of laboratory methods and sequencing strategies precluded a standardized workflow. In contrast, the 184 ancient unassembled genomes were retained given the relatively standardized, albeit specialized, laboratory procedures required to process ancient tissues.

Collection location, date, and host metadata were curated by cross-referencing the original publications. Locations were transformed to latitude and longitude coordinates using GeoPy (v2.0.0) and the Nominatim API (https://github.com/osm-search/Nominatim) for OpenStreetMap. Coordinates were standardized at the level of country and province/state, using the centroid of each. Collection dates were standardized according to their year and recording uncertainty arising from missing data and radiocarbon estimates. Genomes were removed if no associated date or location information could be identified in the literature, or if there was documented evidence of laboratory manipulation.

Two additional data sets were required for downstream analyses. First, *Y. pestis* strain CO92 (GCA_000009065.1) was used as the reference genome for sequence alignment and annotation. Second, *Yersinia pseudotuberculosis* strains NCTC10275 (GCA_900637475.1) and IP32953 (GCA_000834295.1) served as an outgroup to root the maximum likelihood phylogeny.

**Sequence alignment**. Modern assembled genomes were aligned to the reference genome using snippy (v4.6.0) (https://github.com/tseemann/snippy), a pipeline for core genome alignments. Snippy first shreds contigs into 250 bp single-end reads at a uniform depth of 20x across the genome, followed by mapping and variant calling. Default parameters were used, along with the following minimum thresholds: depth of 10X, base quality of 20, mapping quality of 30, major allele frequency of 0.9. Modern genomes were excluded if the number of sites covered at a minimum depth of 10X was less than 70% of the reference genome. After applying this filter, 540 modern genomes remained.

Ancient unassembled genomes were downloaded from the SRA database in FASTQ format using the SRA Toolkit (v2.10.8). Pre-processing and alignment to the reference genome was performed using the nf-core/eager pipeline (v2.2.1), a reproducible workflow for ancient genome reconstruction[60]. Default parameters were used, along with the following minimum filters: read length of 35 bp, an edit distance of 0.01, and a 16 bp seed length. Only merged reads were retained from paired end-sequencing projects. Ancient genomes were removed if the number of sites covered at a minimum depth of 3X was less than 70% of the reference genome. After applying this filter, 61 ancient genomes remained.

A multiple sequence alignment was constructed using the snippy core module of the snippy pipeline (v4.6.0). The output alignment was filtered to only include chromosomal sites that were present in at least 95% of samples (i.e., a missing data threshold of 5%). The filtered alignment included 10,249 variant positions exclusive to *Y. pestis*, with 6405 variant positions found in only a single strain and 3844 sites shared by at least two strains.

**Maximum likelihood phylogenetic analysis**. Model selection was performed on the full data set ($N = 601$) using Modelfinder[61] as implemented in IQTREE2 (v2.2.1). Modelfinder identified the K3Pu + F + I model as the optimal choice based on the Bayesian Information Criterion (BIC). The K3P model, also known as K81, estimates substitution rates using three categories, in this case: (1) A < - > C equals G < - > T, (2) A < - > G equals C < - > T, and (3) A < - > T equals C < - > G). The "u + F" suffix indicates that base frequencies will be empirically evaluated and thus are not assumed to be equal. The "+I" suffix indicates that a proportion of the alignment includes invariable sites (i.e., non-SNPS),

A maximum likelihood phylogeny was estimated for this data across 10 independent runs of IQTREE2 (v2.2.1)[62]. Branch support was evaluated using 1000 iterations of the ultrafast bootstrap approximation[63], with a threshold of 95% required for strong support. All phylogenetic data, visualizations are publicly accessible and interactive at: https://nextstrain.org/community/ktmeaton/yersinia-pestis

**Statistics and reproducibility**. Each subsection of the methods contains detailed explanations of various statistical models used to test the veracity of the data presented in this paper, including multiple replications of subsampled data and tree robustness.

**Data partitions**. The full multiple sequence alignment was alternatively split into 12 populations, referred to as the population data sets. These populations were defined by the intersection of the following nomenclature systems: the major branches, metabolic biovars, and historical time period (Table S1). One sample was excluded, a *Pestoides* isolate from the Bronze Age (Strain RT5, BioSample Accession SAMEA104488961), as this population would be of size $N = 1$.

In an attempt to improve the performance and convergence of molecular clock analyses, a subsampled data set was also constructed. Populations that contained multiple samples drawn from the same geographic location and the same time period were reduced to one representative sample. The sample with the shortest terminal branch length was prioritized, to diminish the influence of uniquely derived mutations on the estimated substitution rate. An interval of 25 years was identified as striking an optimal balance, resulting in 191 samples, which is a 68% reduction from the original data set.

**Estimating rates of evolutionary change**. To explore the degree of temporal signal present in the data, two categories of tests were performed. The first was a root-to-tip (RTT) regression on the mean sampling age using the *statsmodels* python package. Given the relative simplicity of a regression model, the full data set of 601 genomes was used. We also used RTT regressions to visualize temporal signal for each population. In cases where temporal signal was weak, for example where the R2 was very low or the slope was negative, we attempted to remove the top 10% longest branches, which did not improve the regressions, such that we decided to keep the complete data sets for further analyses to avoid arbitrary selection of data.

For the second test of temporal signal, a Bayesian Evaluation of Temporal Signal (BETS) was conducted on the full dataset as well as separately by population. This consisted of running four model configurations: either with or without sampling dates, and under a strict or uncorrelated lognormal relaxed clock model (strict and UCLN, respectively). We calculated the log marginal likelihood under each model configuration using stepping-stone sampling as implemented in BEAST (v1.10)[64]. To this end, we ran 200 path steps, each with a Markov chain Monte Carlo (MCMC) of length $10^6$ steps (Supplementary Table S3). In addition to the clock model, we used a GTR + gamma nucleotide substitution model. We chose a different substitution model from that selected for maximum-likelihood inference as Bayesian analyses are more robust to mild over-parameterization, and thus we chose GTR, of which K3P is nested within. Given enough phylogenetic signal in the data, and the absence of any population-specific variation, the Bayesian estimates of the substitution rate matrix is expected to approximately converge to the K3P model.

Bayesian phylogenetic analyses require a tree prior, for which we set a constant-size coalescent. Clearly, we anticipate our data to display some population structure and fluctuations in population size over time. Our choice of the constant-size coalescent is therefore for statistical convenience, rather than it being an accurate description of the demographic process. In particular, the constant-size coalescent has a single parameter, the population size, for which it is easy to specify a proper prior, unlike some skyline models, which is essential for marginal likelihood calculations. Moreover, because our data have many variable sites, we expect them to be highly informative and therefore override the signal from the tree prior for estimates of evolutionary rates and times[65,66].

Importantly, the models involved priors that were proper for all parameters, which is essential for marginal likelihood calculations[67]. In particular, the molecular clock rate (i.e., the mean of the UCLN clock model or the global rate of the strict clock) had a continuous time Markov chain reference prior[68], the population size of the constant-size coalescent an exponential prior distribution with mean 10, and the standard deviation of the UCLN had an exponential prior with mean 0.33. Marginal likelihood estimation with stepping-stone sampling does not require the posterior distribution. To obtain the posterior distribution we used an MCMC of $10^9$ steps, sampling every $10^3$ steps. For situations where the effective sample size (ESS) of any parameters was below 200, we increased the chain length by 50% and reduced sampling frequency accordingly.

**Estimating ancestral locations and spread**. To explore underlying phylogeography, we performed ancestral state reconstruction using the maximum likelihood method implemented in TreeTime (v0.8.1)[51]. We independently fit three discrete mutation models to the maximum likelihood phylogeny using the sampling locations by: (1) continent, (2) country, and (3) province. The mapping of

countries to continents was defined according to the open-source resource GeoJSONRegions (https://geojson-maps.ash.ms/). For each internal node, we extracted the ancestral location with the highest likelihood given the data.

We also conducted a discrete trait analysis in BEAST (v1.10)[64,69]. Country of sample origin was chosen as the discrete trait of interest. A coalescent constant population size tree prior was chosen with an exponential prior placed on the effective population size with mean 100000. We modeled evolutionary rate with an uncorrelated relaxed lognormal clock, with a CTMC scale prior on the mean and exponential prior with mean 1/3 on the standard deviation of the underlying lognormal distribution[70]. A GTR + gamma nucleotide substitution model with estimated base frequencies for 1.ORI, 1.PRE, 0.ANT4, and 0.PRE. The same settings were used for 2.MED with the exception of swapping the GTR + gamma model to an HKY + gamma model. MCMC chains were run for $10^7$ steps with sampling every $10^3$ steps. We used logCombiner to combine between 3–5 replicate runs, with 10% burnin, for each clade to achieve ESS above 200 for each parameter and Maximum Clade Credibility (MCC) trees[71].

**Visualization**. Data visualization was performed using the python package seaborn (v0.11.1)[72] and Auspice (v2.23.0)[73], a component of the Nextstrain visualization suite.

**Reporting summary**. Further information on research design is available in the Nature Portfolio Reporting Summary linked to this article.

## Data availability

All genomic sequences used in this study are publicly available and were downloaded from the National Center for Biotechnology Information (https://www.ncbi.nlm.nih.gov/). Genomic metadata and accession numbers are described in Supplementary Table S8. All phylogenetic trees, molecular clock analyses, and geographic reconstructions are publicly available as interactive Nextstrain exhibits: https://nextstrain.org/community/ktmeaton/yersinia-pestis

## Code availability

The computational workflow is publicly available as a snakemake pipeline (https://github.com/ktmeaton/plague-phylogeography/).

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

## Acknowledgements

This work was supported by the Social Sciences and Humanities Research Council of Canada (#20008499), CIFAR humans and the microbiome program (HNP), and the MacDATA Institute (McMaster University, Canada). This research was enabled in part by support provided by Compute Ontario (https://www.computeontario.ca/) and Compute Canada (https://www.computecanada.ca). We would like to thank Madeline Tapson, Dr. Dan Salkeld, and Dr. Jennifer Klunk for their expertise in contextualizing the ecology and evolutionary history of plague. We also thank Jessica Hider and Marie-Hélène B.-Hardy for discussions on the interpretation of genomic data from zoonotic pathogens. We are indebted to Dr. Ana Duggan and Dr. Emil Karpinski for their insight regarding Bayesian methods for phylogenetic analysis. We thank members of the Sherman Center for Digital Scholarship, including Dr. Andrea Zeffiro, Dr. John Fink, Dr. Matthew Davis, and Dr. Amanda Montague, for their assistance in developing the genomic database. Finally, we would like to thank Debi Poinar and all past and present members of the McMaster Ancient DNA Center and the Golding Lab at McMaster University as well as three anonymous reviewers for their insightful and constructive comments.

## Author contributions

K.E., G.B.G., and H.N.P. designed the study. K.E., L.F., and S.D. performed computational analysis. A.G.C. and N.V. provided historical sources and interpretation. E.C.H. and HP critiqued the computational methods and discussion. G.B.G. provided access to computational resources and data storage. H.N.P. and G.B.G. supervised the study. K.E. wrote the manuscript with contributions from all co-authors.

## Competing interests

The authors declare no competing interests.
