## [Peer Review File · Communications Biology]

Reviewers' comments:

Reviewer #1 (Remarks to the Author):

Through published genomic data, the authors re-analyzed the population specific evolutionary rates of *Y. pestis*, and evaluated the justification of phylogeographical analysis in *Y. pestis* using solely genomic evidences. The improved estimation on divergent time of key nodes on genealogy would help to correlate the historical events with evolution of *Y. pestis* in future studies.

Major comments:

1. There is strong temporal signal presented in 0.PRE population, of which sampling time period lasted for ~1500 years, however, in the similar time scaffold, the temporal signal disappeared in the whole species level. Could the authors discuss the possible reason on this?
2. There are lot of strains revealed ultra long branch length (grey branch in the phylogeny). Are there any possible that these long branch strains were caused by poor assembly quality, and how can these outliers influence the estimation of rate variations?

Minor comments:

1. Did the author include strain of 0.PE3 in the analysis?
2. Table 1, it would help a lot if the authors clarify the rule on definition of different categories.
3. Fig. 5, I like this plot-very informative, but could the author add a panel that use all genomes (including ones from human), as control?
4. Fig. 6, it is difficult to distinguish the color of 1.ORI and 1.IN.
5. Fig. 7, the legend, 0.PRE should be 1.PRE.
6. Method, Sequence Alignment section, as author claimed they only used assembled genome in the analysis, it is quite weird to emphasize parameters like "depth of 10X, base quality of 20, mapping quality of 30, major allele frequency of 0.9".

Reviewer #2 (Remarks to the Author):

[This review is also included as an attachment for easier viewing]

In this study, entitled "Plagued by a cryptic clock: Insight and issues from the global phylogeny of *Yersinia pestis*", Eaton and colleagues explore published *Y. pestis* genomic diversity to assess substitution rate variations across different phylogenetic lineages and their impact on the global clock rate. Using a published dataset of ~600 genomes, the authors identify an absence of temporal signal across the entire *Y. pestis* phylogeny and rather propose a lineage-specific approach for molecular dating. Moreover, the authors identify sufficient phylogenetic signal in five "populations" that encompass wider temporal sampling ranges. Individual molecular dating analyses were then used to re-estimate the dates of origin for the respective "populations" and to discuss possible historical implications of their inferred divergence dates. Finally, the authors explore possible dissemination patterns in relation to historical epidemics/pandemics, and identify significant migration events during the Third Plague Pandemic, which coincide with documented historical accounts.

While I do appreciate the authors effort towards a comprehensive characterisation of all *Y. pestis* clades using a methodology that has not been presented before, I have a number of major and minor comments regarding the analytical approach and interpretation of results. Moreover, I find that relevant supporting display items are missing for fully assisting the authors arguments. My specific comments can be found below.

Major points:

- The authors should provide a table of all 601 genomes used in this study together with their accession codes, isolation dates, geographic coordinates and isolation hosts in order to ensure transparency in the used dataset and aid reproducibility of the analysis.

- Is there a specific reason why the oldest documented *Y. pestis* lineages were not included in this study? Currently, they are represented by the genomes Gok2 or RV2039. I suggest their addition as they might help improve the phylogenetic signal across all *Y. pestis*.
- One of the determining priors in Bayesian molecular dating analysis is the demographic model used. Yet, no testing of demographic models is performed here and the authors choose to proceed with a coalescent constant population size model that seems inappropriate for describing the evolutionary history of epidemic pathogens. The suitability of other models, such as the coalescent skyline and exponential growth models should be fully evaluated for the entire dataset, as well as all the subsets.
- The authors use different substitution models for the maximum likelihood and Bayesian phylogenetic analyses. If no specific reasoning exists, the model chosen by ModelFinder should be used for all analyses.
- The first section of this paper "Population Structure" includes a critical discussion of genomic nomenclature and population distinctions in *Y. pestis*. Yet the authors choose to use those arbitrary distinctions between populations for their subsequent analyses. How are the 12 "populations" defined in this study? Given their criticism regarding the distinction between the second and third pandemic, could all post-Big Bang lineages be considered as one population? Is a molecular dating analysis of combined Branches 1-4 supported by BETS?
- The authors mention that "BETS was inconclusive when attempting to fit a single clock to the updated global diversity of *Y. pestis*" but the accompanying Bayes Factor comparisons are not presented in the paper. To aid transparency, those should be presented as a supplementary display item.
- To what extent could false SNPs affect estimates of rate variation across different phylogenetic lineages? For example, I notice very long terminal branch lengths in some 2.MED and 0.PE genomes, which are probably responsible for skewing the clock rate on these lineages. How are possible artefactual variants (SNPs) evaluated and accounted for in this study?
- The authors mention that "The inability to infer divergence dates due to sampling bias also has several historical implications." Could informative dates be deduced from 0.ANT lineages when co-analysed with their ancient representatives? The possibility of such an analysis should be evaluated with BETS. Moreover, the authors mention, "Perhaps the most significant concerns the emergence of plague in Africa which makes up 90% of all modern plague cases, yet for which there remains not a single ancient sequence." Given the inability to estimate realistic dates for 1.ANT alone, could these be better estimated when considering all Branch 1 genomes as a single group? Is such a setup supported through model-testing (BETS approach)?
- Within the section "Informative Rates and Dates" the resulting maximum clade credibility trees from all analyses should be presented. Moreover, the point of "origin" in each case should be indicated on the trees with its corresponding 95% HPD interval.
- Regarding the estimated 1214-1315 date for the origin of 1.PRE, what is the genetic distance of the most basal genome on this lineage from the "point of origin"? In Figure S6, it seems that the MRCA of 1.PRE is placed at a zero genetic distance from the most basal genome. What is the isolation location and archaeological date of this genome? Based on the cited literature, the earliest Second Plague Pandemic genomes are dated to the 14th

century. If the observation of zero genetic distance to the MRCA is correct, what does an origin date of 1214 mean and is it realistic? Given the important historical implications discussed by the authors, I emphasize that feasibility should not be confused with statistical uncertainty and that the resulting dates should be discussed in a more balanced and critical manner in the Results and Conclusions sections.

- Similarly to the comment above, what is the genetic distance of the most basal genome from the MRCA of the 0.ANT4 lineage? Given the estimate of one mutation every ~9.5 years for *Y. pestis*, how realistic is the result of a 272 CE origin date? In terms of interpretation, does the origin of a pandemic always coincide with the origin of its associated lineage? Again, the authors should discuss these aspects in a more balanced and critical manner within the Results and Conclusions sections.
- Other studies using different approaches have reached comparable results when estimating the date intervals for the emergence of the First and Second Pandemic lineages. Such studies should be credited within the Results and Conclusions sections.
- The authors mention "...we independently fit three discrete migration models to the maximum likelihood phylogeny using the sampling locations by: (1) continent, (2) country, and (3) province" Should time not be also considered as a factor for these migration models?
- Migration analysis: Why were the First Pandemic and the Bronze Age lineages not considered for this analysis? The latter might be a particularly good candidate as it shows an almost perfect temporal signal and is sampled from diverse locations.
- The authors mention "We estimated that the Second Pandemic (1.PRE) diverged from an ancestral population located in China (probability: 0.93) as part of the "Big Bang" polytomy." This statement is rather confusing as to what exactly it means. Figure 7 shows that it was the 0.ANT lineage that diverged in China, not the 1.PRE lineage. Also, how is the 1.PRE divergence separated from the beginning of the Second Pandemic? Are those not referring to the same divergence? Given the low support of these models, a clear description is important here. Moreover, what do the yellow arrows in Figure 7 indicate and why are they not described within the corresponding results section? If these are inferred migration events of higher support then I would suggest they are appropriately discussed. Alternatively, the authors should give a reasoning why these results cannot be trusted. If the results cannot be trusted with regard to migration events, but are in other ways informative for the study, I would suggest this entire analysis is moved to the supplement.

Minor points:

- "The most intensively researched events have been: (1) the first appearance of *Y. pestis* in human populations" This statement includes an incorrect citation and should be updated with more recent studies.
- "(3) the inter-pandemic or "quiescent" periods where *Y. pestis* recedes into wild rodent reservoirs and disappears from the historical record" Ancient *Y. pestis* evidence from the 3rd century AD could be included here, as it is a good example of a historical genome that does not fall within the defined "pandemic" periods.
- "As a result, considerable debate has emerged over whether *Y. pestis* has no temporal signal" Here only one citation is included. Please include more to highlight different

views on the debate.

- “This uncertainty has resulted in radically different rate and date estimates between studies, with node dates shifting by several millennia.” Here the addition of basal lineages contributing to a tightening of the estimated emergence dates should be included as a factor.
- “This contention concerns competing hypotheses about the relative importance of localized persistence versus long-distance reintroduction.” This sentence is presented in the context of the pandemic’s origins, yet it seems more relevant to the pandemic’s progression. This precise association is not clear here and should be further clarified.
- “Among both sides of this issue, there is an expectation that genomic evidence will play a significant role, if not resolve the debate” The indicated citations within this sentence seem inappropriate as they do not present new genomic evidence or help resolved the debate regarding the pandemic’s origin.
- It is unclear what is meant by “with 3,844 sites shared by at least two strains”. What does this information contribute?
- Regarding the observation “The Pestoides (0.PE) and Medievalis (2.MED) biovars are informative examples, as these populations have co-existed in the Caucasus mountains since at least the 20th century”, are these different biovar perhaps identified in distinct hosts? In this case a table of isolation hosts for all the presented strains would be appreciated.
- “Phylogenetic analysis reveals genetic continuity between these two events, as the Third Pandemic (1.ORI) is a direct descendant of the Second Pandemic (1.PRE).” While this statement is true, one of the major lineages responsible for the Second Plague Pandemic, is one that currently includes the majority of data produced from this period and is today extinct. In that regard, the consideration of Second and Third pandemic strains as being genetically distinct, at least to an extinct, is not completely arbitrary and should be mentioned here.
- Figure 1: A higher resolution phylogeny would be useful as a supporting display item, where the phylogenetic positioning of each individual strain, with its respective name is shown.
- Figure 1: A Bronze Age genome associated with 0.PE is missing from the “Time Period” part of panel A.
- Figure 1: Why is the polytomy shown as a separate event from the beginning of the pandemic? An explanation should be provided.
- Figure 4 is a bit confusing as to what the boxes with dates indicate. A temporal indication of all depicted strains would make the figure clearer.
- Figure 5: An accompanying table with isolation hosts from all genomes used for this paper should be included.
- Figure 6+7: The migration analysis maps shown here should be presented with their corresponding phylogenetic trees, to aid visualisation and interpretation.
- Figure 6+7: I suggest a removal of all non-significant migration arrows from the main figures as they can be confusing and lead to false interpretations.

- There is a typo in the legend of Figure 7, 0.PRE should be 1.PRE.
- There is a typo in the Conclusions section "Plague of Justinian (531CE)" should be 541.

Reviewer #1

Expertise

Pathogen genomics, *Y. pestis*

Comments

Through published genomic data, the authors re-analyzed the population specific evolutionary rates of *Y. pestis*, and evaluated the justification of phylogeographical analysis in *Y. pestis* using solely genomic evidences. The improved estimation on divergent time of key nodes on genealogy would help to correlate the historical events with evolution of *Y. pestis* in future studies.

Major

1. There is strong temporal signal presented in O.PRE population, of which sampling time period lasted for ~1500 years, however, in the similar time scaffold, the temporal signal disappeared in the whole species level. Could the authors discuss the possible reason on this?

*The mechanisms that govern rate variation within *Y. pestis* are currently not well understood. The two major hypotheses are (1) adaptations to new environments and hosts, and (2) changing bacterial replication rates between epidemic and endemic cycles (Cui et al., 2013). To date, little support for either hypothesis has been found using genomic data. Our paper focuses on an earlier stage of this analysis, which is obtaining more reliable estimates of the rate variation, and critiquing their interpretation and application. We hope this manuscript will serve as a foundation for future work to test which variables explain (or do not explain) the population-specific patterns observed here.*

2. There are lot of strains revealed ultra long branch length (grey branch in the phylogeny). Are there any possible that these long branch strains were caused by poor assembly quality, and how can these outliers influence the estimation of rate variations?

*We have added a new table **Dataset 1: Supplementary Table S9** to characterize the relationship between sequencing technology, assembler, and long branch outliers. We did not observe any strong patterns, although we note that long branches were most frequently associated with IonTorrent data assembled with Newbler (23 sequences) and Illumina data assembled with SPAdes (17 sequences).*

*Poor assembly quality can inflate estimates of biological rate variation due to the presence of artifactual SNPs (sequencing/assembly error). However, sequence curation of *Y. pestis**

assemblies is challenging as the raw reads (fastq) are often not publicly available, or have been generated with such diverse library prep methods that bioinformatic analysis must be tailored to each individual dataset. Exploring whether assembly quality could be retroactively improved would be a valuable experiment, but is outside the scope of this manuscript.

Minor

1. Did the author include strain of 0.PE3 in the analysis?

0.PE3, Angola, was not used in this study as it's exceptionally long branch length has been previously observed to interfere with molecular clock dating and prevent Bayesian analyses from converging (Cui et al., 2013).

2. Table 1, it would help a lot if the authors clarify the rule on definition of different categories.

*The caption to **Table 1** has been extended to include definitions for each category. In addition, categories were re-named to match the in-text paragraph header where they are discussed in detail.*

3. Fig. 5, I like this plot-very informative, but could the author add a panel that use all genomes (including ones from human), as control?

*The data underlying **Figure 5** includes all genomes used in this study (N=601, including humans) grouped by population. The plotting data is available in **Dataset 1: Supplementary Table S6**. If we have misunderstood your request, would you be willing to provide additional explanation?*

4. Fig. 6, it is difficult to distinguish the color of 1.ORI and 1.IN.

Thank you for this suggestion, we agree that the colors are nearly indistinguishable. We have adjusted Figure 6 to use shapes to distinguish 1.ORI and 1.IN (square vs. circle) and have lightened the shade of 1.IN in this plot to further improve contrast.

5. Fig. 7, the legend, 0.PRE should be 1.PRE.

We thank the reviewer for catching this and have adjusted the legend of Figure 7.

6. Method, Sequence Alignment section, as author claimed they only used assembled genome in the analysis, it is quite weird to emphasize parameters like "depth of 10X, base quality of 20, mapping quality of 30, major allele frequency of 0.9".

To align assemblies to the reference, we use the snippy pipeline which first shreds contigs into 250 bp single-end reads before using a short-read aligner (BWA) for mapping. We have added this explanation to the section Materials and Methods: Sequence Alignment.

Reviewer #2

Expertise

Archaeogenetics, phylogeography of *Y. pestis*

Comments

In this study, entitled “Plagued by a cryptic clock: Insight and issues from the global phylogeny of *Yersinia pestis*”, Eaton and colleagues explore published *Y. pestis* genomic diversity to assess substitution rate variations across different phylogenetic lineages and their impact on the global clock rate. Using a published dataset of ~600 genomes, the authors identify an absence of temporal signal across the entire *Y. pestis* phylogeny and rather propose a lineage-specific approach for molecular dating. Moreover, the authors identify sufficient phylogenetic signal in five “populations” that encompass wider temporal sampling ranges. Individual molecular dating analyses were then used to re-estimate the dates of origin for the respective “populations” and to discuss possible historical implications of their inferred divergence dates. Finally, the authors explore possible dissemination patterns in relation to historical epidemics/pandemics, and identify significant migration events during the Third Plague Pandemic, which coincide with documented historical accounts.

While I do appreciate the authors effort towards a comprehensive characterisation of all *Y. pestis* clades using a methodology that has not been presented before, I have a number of major and minor comments regarding the analytical approach and interpretation of results. Moreover, I find that relevant supporting display items are missing for fully assisting the authors arguments. My specific comments can be found below.

Major

1. The authors should provide a table of all 601 genomes used in this study together with their accession codes, isolation dates, geographic coordinates and isolation hosts in order to ensure transparency in the used dataset and aid reproducibility of the analysis.

*Genomic metadata is provided in **Dataset 1: Supplementary Table S8**.*

2. Is there a specific reason why the oldest documented *Y. pestis* lineages were not included in this study? Currently, they are represented by the genomes Gok2 or RV2039. I suggest their addition as they might help improve the phylogenetic signal across all *Y. Pestis*.

*Gok2 (Rascovan et al., 2019) was excluded as it did not meet our coverage filter of 70% of the genome covered at a minimum depth of 3X. RV2039 (Susat et al., 2021) was not used in this study as it was published on 2021 June 29, which post-dated our data collection period (2020 January 01). We have updated the section **Methods: Data Collection** to clarify our data collection cut-off.*

3. One of the determining priors in Bayesian molecular dating analysis is the demographic model used. Yet, no testing of demographic models is performed here and the authors choose to proceed with a coalescent constant population size model that seems inappropriate for describing the evolutionary history of epidemic pathogens. The suitability of other models, such as the coalescent skyline and exponential growth models should be fully evaluated for the entire dataset, as well as all the subsets.

*This is an important point and we have explained our reasoning for our choice of tree prior in section **Methods: estimating rates of evolutionary change**.*

We do agree with the reviewer that the constant coalescent tree prior is overly simplistic for these data. However, the only models appropriate here are those that account for population structure (e.g. structured coalescent or multitype birth-death), which are computationally prohibitive for our data. For this reason we chose a model that was as simple as possible and expected to have little impact in estimates of node times when the data are informative (see Möller et al., 2018; Ritchie & Ho, 2019). Moreover, we used the most accurate model selection technique, generalised stepping stone, that requires a fully proper prior, which is not possible with the Bayesian skyline.

4. The authors use different substitution models for the maximum likelihood and Bayesian phylogenetic analyses. If no specific reasoning exists, the model chosen by ModelFinder should be used for all analyses.

We used different substitution models for two reasons.

*First, is the data composition differs between the maximum-likelihood phylogeny (all populations combined) and the Bayesian phylogeny (each population modeled separately). For the maximum-likelihood inference, the K3P model was selected by Modelfinder as the best fit for the entire *Y. pestis* species (+ the *Y. pseudotuberculosis* outgroup). For the Bayesian analysis, we used the generalized GTR model, which K3P is nested within, to account for population-specific variation.*

The second reason is that Bayesian analyses are robust to mild over-parameterization and if there is enough signal in the data, we will recover the “true” substitution matrix. Even though a GTR model was specified (all substitutions can occur at different rates), our posterior estimates of the substitution rate matrix approximated the K3P model (AC equals GT, AG equals CT, AT equals CG). For example, here is the Bayesian substitution matrix for 1.PRE:

AC: 1.3 ~ CG: 1.7

AG: 1.1 ~ = CT: 1.2
AT: 0.4 ~ = CG: 0.3

*In summary, although we provided a more flexible model to BEAST (ex. GTR) our posterior estimates converged to the same substitution model (K3P) as used in the maximum-likelihood inference. We have added these additional explanations to the section **Methods: Estimating Rates of Evolutionary Change**.*

5. The first section of this paper “Population Structure” includes a critical discussion of genomic nomenclature and population distinctions in *Y. pestis*. Yet the authors choose to use those arbitrary distinctions between populations for their subsequent analyses. How are the 12 “populations” defined in this study? Given their criticism regarding the distinction between the second and third pandemic, could all post-Big Bang lineages be considered as one population? Is a molecular dating analysis of combined Branches 1-4 supported by BETS?

*We have expanded the section **Population Structure: Integrative Approach** to clarify that the populations are defined by having a unique combination of a major branch, biovar, and time period. Further information about each population is available in **Dataset 1: Supplementary Table S1**.*

We agree with the reviewer that testing temporal signal in combined populations (ex. Branches 1,2,3,4) is an informative avenue for plague research, but at present we have no strong arguments for combining populations for tests of temporal signal. Moreover, our results here are readily comparable between populations and avoid several caveats of analysing multiple populations within a single phylogenetic analyses.

6. The authors mention that “BETS was inconclusive when attempting to fit a single clock to the updated global diversity of *Y. pestis*” but the accompanying Bayes Factor comparisons are not presented in the paper. To aid transparency, those should be presented as a supplementary display item.

*The reviewer makes a valuable point here. In the manuscript we state that we could not achieve reliable sampling from the posterior using MCMC, which means that marginal likelihood calculations and all parameters are unreliable, which is not the same thing as the Bayes factors being inconclusive. To make it clearer we have rephrased the section **Estimating Rates of Evolutionary Change** to better explain this point.*

7. To what extent could false SNPs affect estimates of rate variation across different phylogenetic lineages? For example, I notice very long terminal branch lengths in some 2.MED and 0.PE genomes, which are probably responsible for skewing the clock rate on these lineages. How are possible artefactual variants (SNPs) evaluated and accounted for in this study?

*We have been especially careful to exclude artifactual data to the greatest extent possible. In fact we've been exceptionally conservative for this exact reason. As such this dataset represents the most carefully curated dataset of Yp sequences to date. To make all this transparent we have added a new table **Dataset 1: Supplementary Table S9** to characterize the relationship between sequencing technology, assembler, and long branch outliers. We did not observe any strong patterns, although we note that long branches were most frequently associated with IonTorrent data assembled with Newbler (23 sequences) and Illumina data assembled with SPAdes (17 sequences).*

*Poor assembly quality can inflate estimates of biological rate variation due to the presence of artifactual SNPs (sequencing/assembly error). However, sequence curation of *Y. pestis* assemblies is challenging as the raw reads (fastq) are often not publicly available, or have been generated with such diverse library prep methods that bioinformatic analysis must be tailored to each individual dataset. Exploring whether assembly quality could be retroactively improved would be a valuable experiment, but is outside the scope of this manuscript.*

*Artefactual variants are one possible explanation for the inflated evolutionary rate estimates in 2.MED and 0.PE. However, the relationship between the proportion of long branches and the rate variation in a population is also unclear. The populations with the greatest number of long branches (*pestoides* 0.PE and *medievalis* 2.MED) both had detectable temporal signal according to BETS. But of the two, informative rates and divergence dates could only be obtained from *medievalis* 2.MED. This suggests that numerous long-branches (possibly from artifactual SNPs) does not necessarily preclude recovery of temporal signal.*

8. The authors mention that “The inability to infer divergence dates due to sampling bias also has several historical implications.” Could informative dates be deduced from 0.ANT lineages when co-analysed with their ancient representatives? The possibility of such an analysis should be evaluated with BETS. Moreover, the authors mention, “Perhaps the most significant concerns the emergence of plague in Africa which makes up 90% of all modern plague cases, yet for which there remains not a single ancient sequence.” Given the inability to estimate realistic dates for 1.ANT alone, could these be better estimated when considering all Branch 1 genomes as a single group? Is such a setup supported through model-testing (BETS approach)?

*We agree that experimenting with population definitions (combining/splitting) is the intuitive next step in improving molecular clock analyses in *Y. pestis*. Similar to our response to Reviewer 2's Major point #5, however the complexity of the experimental design and scale of that analysis (i.e. testing all possible population combinations) is outside the scope of this work and would be better served as a standalone publication.*

9. Within the section “Informative Rates and Dates” the resulting maximum clade credibility trees from all analyses should be presented. Moreover, the point of “origin” in each case should be indicated on the trees with its corresponding 95% HPD interval.

We have created a new supplementary dataset (**Dataset 2**) which contains tree files (newick and nexus) as well as a formatted metadata file (tsv) for visualization. In addition, we have created a <https://nextstrain.org/> community page to host interactive versions of all of our phylogenies: <https://nextstrain.org/community/ktmeaton/yersinia-pestis>

10. Regarding the estimated 1214-1315 date for the origin of 1.PRE, what is the genetic distance of the most basal genome on this lineage from the “point of origin”? In Figure S6, it seems that the MRCA of 1.PRE is placed at a zero genetic distance from the most basal genome. What is the isolation location and archaeological date of this genome? Based on the cited literature, the earliest Second Plague Pandemic genomes are dated to the 14th century. If the observation of zero genetic distance to the MRCA is correct, what does an origin date of 1214 mean and is it realistic? Given the important historical implications discussed by the authors, I emphasize that feasibility should not be confused with statistical uncertainty and that the resulting dates should be discussed in a more balanced and critical manner in the Results and Conclusions sections.

*The most basal genome of the Second Pandemic (1.PRE) is strain LAI009 from Russia with a collection date interval of 1300:1400. The divergence from the MRCA is 2.36×10^{-8} substitutions/site * 4,229,098 sites = effectively 0 substitutions. From our Bayesian analysis, the median substitution rate for 1.PRE was 9.51 years/substitution (**Dataset 1 : Supplementary Table S4**). Hypothetically, if the collection date was truly on the lower bound (1300 CE), it is possible that the MRCA could be 9.5 years before that (1290 CE). In this rough thought experiment, 1290 CE is contained in the confidence interval of 1214-1315 CE and is plausible (i.e. the branch lengths in the maximum-likelihood tree are congruent with the Bayesian estimates).*

*However, a larger methodological concern is uncertainty in estimating the root position when only samples from within a population are used (no outgroup). We have included the maximum-clade-credibility tree of the Second Pandemic (1.PRE) as **Supplementary Figure S7**. We have included an extra paragraph in the section **Estimating Rates of Evolutionary Change: Informative Rates and Dates** to expand on methodological issues that might impact our date estimates, and refer back to this in the conclusion.*

11. Similarly to the comment above, what is the genetic distance of the most basal genome from the MRCA of the 0.ANT4 lineage? Given the estimate of one mutation every ~9.5 years for *Y. pestis*, how realistic is the result of a 272 CE origin date? In terms of interpretation, does the origin of a pandemic always coincide with the origin of its associated lineage? Again, the authors should discuss these aspects in a more balanced and critical manner within the Results and Conclusions sections.

*If we exclude DA101 (3rd century CE), the most basal strain of the First Pandemic (0.ANT4) is EDI003 from England with a collection date interval of 500:650. The divergence from the MRCA is 2.02×10^{-6} substitutions/site * 4,229,098 sites = 8.54 substitutions. From our Bayesian analysis, the median substitution rate for 0.ANT4 was 13.36 years/substitution (**Dataset 1 : Supplementary Table S4**). Hypothetically, if the collection date was truly on the lower bound*

(500 CE), it is possible that the MRCA could be as early as 114 years (8.54 substitutions * 13.36 years/substitution) before that (386 CE). In this rough thought experiment, 386 CE is contained in the confidence interval of 272:465 CE and is plausible.

Unlike the Second Pandemic (1.PRE) the root location of the First Pandemic is estimated with high confidence in both the maximum-likelihood and maximum-clade-credibility tree. Thus, there is higher confidence in the estimate dates of the tMRCA. We have included the maximum-clade-credibility tree of the First Pandemic (0.ANT4) as **Supplementary Figure S8**.

12. ✓ Other studies using different approaches have reached comparable results when estimating the date intervals for the emergence of the First and Second Pandemic lineages. Such studies should be credited within the Results and Conclusions sections.

We have added additional tMRCA estimates to **Table 1** where they have been reported (Rasmussen et al., 2015; Spyrou et al., 2019). However, two studies (Namouchi et al., 2018), (Spyrou et al., 2018) do not provide dates. Namouchi et al. don't report any time estimates due to a lack of temporal signal in their dataset and Spyrou et al. 2019, only report the estimate of the origin of all plague, which we didn't not attempt and as such it is not directly comparable. We have left these as NA in the Table 1.

13. ✓ The authors mention "...we independently fit three discrete migration models to the maximum likelihood phylogeny using the sampling locations by: (1) continent, (2) country, and (3) province" Should time not be also considered as a factor for these migration models?

We agree that time is an important factor and is the preferred approach (i.e. jointly estimating the phylogeny, molecular clock, and geographic location). However, as we demonstrate here, determining a molecular clock model for Plague is difficult due to dramatic variation in the evolutionary rate and population dynamics. By avoiding a molecular clock model our approach effectively results in an 'overdispersed clock', where each branch is allowed an independent rate. We argue that this is the most conservative approach for phylogeography of this organism, until a more realistic molecular clock model is defined. We explained this point in the corresponding section: "We chose this approach, over one where time is coestimated (i.e. a molecular clock) because the ages of some samples have uncertainties associated with them and due to the large rate variation across the entire tree, which would be a large source of error in migration rates and events over time".

14. ✓ Migration analysis: Why were the First Pandemic and the Bronze Age lineages not considered for this analysis? The latter might be a particularly good candidate as it shows an almost perfect temporal signal and is sampled from diverse locations.

*The migration analysis of the First Pandemic and Bronze Age lineages were not reported on in the manuscript as the results mirror that of the Second Pandemic (i.e. very few migration events can be confidently estimated). This is due to the same issues discussed in the section **Estimating Ancestral Locations and Spread: The Second Pandemic, which** is that the migration rate of plague is often faster than its evolutionary rate. In addition, the First Pandemic and Bronze are even more sparsely sampled than the Second Pandemic both temporal and geographically:*

- Bronze Age/0.PRE: 8 genomes, 4 countries, ~1000 years.
- First Pandemic/0.ANT4: 12 genomes, 5 countries, ~500 years).

The Second Pandemic (1.PRE) is the “best” sampled ancient pandemic and even that had significant problems in the migration analysis. The Third pandemic is not surprisingly the best historically chronicled pandemic. Because we did not generate any unique insights from the analysis of the First Pandemic/Bronze Age, we chose to focus only on the Second Pandemic and Third Pandemic as case studies.

15. The authors mention “We estimated that the Second Pandemic (1.PRE) diverged from an ancestral population located in China (probability: 0.93) as part of the “Big Bang” polytomy.” This statement is rather confusing as to what exactly it means. Figure 7 shows that it was the 0.ANT lineage that diverged in China, not the 1.PRE lineage. Also, how is the 1.PRE divergence separated from the beginning of the Second Pandemic?

This has to do with the fact that what the analysis measures is the support of the stem node (the root of 0.ANT and 1.PRE) vs the crown node, the base of 1.PRE, which is close to the polytomy, but not precisely sitting on it. Figure s1 illustrates the proximity of these two nodes and the clear evidence for ancestral lineages being predominantly based in China. The 1.PRE divergence is estimated back to a MRCA and the confidence around that estimate is given as the 95% CI.

16. Are those not referring to the same divergence? Given the low support of these models, a clear description is important here. Moreover, what do the yellow arrows in Figure 7 indicate and why are they not described within the corresponding results section? If these are inferred migration events of higher support then I would suggest they are appropriately discussed. Alternatively, the authors should give a reasoning why these results cannot be trusted. If the results cannot be trusted with regard to migration events, but are in other ways informative for the study, I would suggest this entire analysis is moved to the supplement.

We now focus our discussion of migration events to those with high statistical support and only arrows with such high support are shown in Figures 6 and 7. We also added some text to explain why we only focus on those migration events with high support: “Importantly, we focus our interpretation of migration events to those that have high statistical support, with high bootstrap (topology inference) and discrete trait reconstruction (phylogeographic inference). In practice, low statistical support in these events means that the data are not

sufficiently informative about migration pathways, and thus their interpretation can be misleading.”

Minor

1. ✓ “The most intensively researched events have been: (1) the first appearance of *Y. pestis* in human populations” This statement includes an incorrect citation and should be updated with more recent studies.

*We have updated the citation of “(1) the first appearance of *Y. pestis* in human populations” to (Susat et al., 2021).*

2. ✓ “(3) the inter-pandemic or “quiescent” periods where *Y. pestis* recedes into wild rodent reservoirs and disappears from the historical record” Ancient *Y. pestis* evidence from the 3rd century AD could be included here, as it is a good example of a historical genome that does not fall within the defined “pandemic” periods.

*We have added a citation to the statement “(3) the inter-pandemic or “quiescent” periods where *Y. pestis* recedes into wild rodent reservoirs and disappears from the historical record” (Damgaard et al., 2018) to reflect the 3rd century strain DA101.*

3. ✓ “As a result, considerable debate has emerged over whether *Y. pestis* has no temporal signal” Here only one citation is included. Please include more to highlight different views on the debate.

*We have updated the citations for this sentence in the **Introduction**, to include four sources that characterize the viewpoints that (1) *Y. pestis* has species-wide temporal signal, (2) *Y. pestis* has population-specific temporal signal, and (3) *Y. pestis* has no temporal signal.*

4. ✓ “This uncertainty has resulted in radically different rate and date estimates between studies, with node dates shifting by several millennia.” Here the addition of basal lineages contributing to a tightening of the estimated emergence dates should be included as a factor.

*We have edited that sentence in the **Introduction** to indicate that divergence dates are shifting and also narrowing, largely due to additional ancient *Y. pestis* genomes.*

5. ✓ “This contention concerns competing hypotheses about the relative importance of localized persistence versus long-distance reintroduction.” This sentence is presented in

the context of the pandemic's origins, yet it seems more relevant to the pandemic's progression. This precise association is not clear here and should be further clarified.

*We have edited the top sentence of that paragraph to "The geographic origins and **progression** of past pandemics are similarly contentious..." to expand the context beyond just the pandemic's origins.*

6. ✓ "Among both sides of this issue, there is an expectation that genomic evidence will play a significant role, if not resolve the debate" The indicated citations within this sentence seem inappropriate as they do not present new genomic evidence or help resolved the debate regarding the pandemic's origin.

*We have adjusted the positioning of the citations within that **Introduction** sentence. We hope to convey that it is not that the cited authors have solved the debate, but rather that they have stated in their articles that they hope/expect additional genomic data to solve it in the future.*

7. ✓ It is unclear what is meant by "with 3,844 sites shared by at least two strains". What does this information contribute?

*We have added additional content to **Methods: Sequence Alignment** to further clarify the variant site composition (3,844 shared variants, 6,405 singleton variants). This information may be useful for readers investigating the degree of genetic diversity in *Y. pestis* (number of SNPs), and how it changes with additional genomes.*

8. ✓ Regarding the observation "The Pestoides (0.PE) and Medievalis (2.MED) biovars are informative examples, as these populations have co-existed in the Caucasus mountains since at least the 20th century", are these different biovar perhaps identified in distinct hosts? In this case a table of isolation hosts for all the presented strains would be appreciated.

*Information about isolation hosts can be found in **Dataset 1: Supplementary Table S8** in the fields `host_raw`, `host_human`, `host_order`. Unfortunately, host information could only be standardized to the level of order due to ambiguity in the available metadata (ex. "rat": Rodentia, "rabbit": Lagamorph).*

9. ✓ "Phylogenetic analysis reveals genetic continuity between these two events, as the Third Pandemic (1.ORI) is a direct descendant of the Second Pandemic (1.PRE)." While this statement is true, one of the major lineages responsible for the Second Plague Pandemic, is one that currently includes the majority of data produced from this period and is today extinct. In that regard, the consideration of Second and Third pandemic strains as being genetically distinct, at least to an extinct, is not completely arbitrary and should be mentioned here.

We have updated the section **Population Structure: Time Period** to clarify that the diversity of Second Pandemic *Y. pestis* (1.PRE) includes both ancestors of the Third Pandemic (1.ORI) as well as uniquely derived lineages.

10. ✓ Figure 1: A higher resolution phylogeny would be useful as a supporting display item, where the phylogenetic positioning of each individual strain, with its respective name is shown.

A high-resolution phylogeny with tips labelled as "Country (Date) Strain" has been included as **Supplementary Figure S9**. An interactive version is also available at: <https://nextstrain.org/community/ktmeaton/yersinia-pestis/maximum-likelihood/all>

11. ✓ Figure 1: A Bronze Age genome associated with 0.PE is missing from the "Time Period" part of panel A.

Thank you for catching this error. We have updated Figure 1: Panel A to properly label the time period of the Bronze Age 0.PE genome.

12. ✓ Figure 1: Why is the polytomy shown as a separate event from the beginning of the pandemic? An explanation should be provided.

The maximum-likelihood phylogeny displayed in Figure 1: Panel A, is a strictly bifurcating tree with no true polytomies. Because of this, the internal node representing the MRCA of Branches 1,2,3 and 4 is distinct from the MRCA of Second Pandemic strains (yellow star). We have updated the caption of Figure 1 to clarify the tree is strictly bifurcating.

13. ✓ Figure 4 is a bit confusing as to what the boxes with dates indicate. A temporal indication of all depicted strains would make the figure clearer.

We have removed the date boxes from Figure 4 as the focus of this

14. ✓ Figure 5: An accompanying table with isolation hosts from all genomes used for this paper should be included.

Information about isolation hosts can be found in Supplementary Table S8 of Dataset 1 in the fields `host_raw`, `host_human`, `host_order`.

15. ✓ Figure 6+7: The migration analysis maps shown here should be presented with their corresponding phylogenetic trees, to aid visualisation and interpretation.

We pointed the reader back to Figure 1, which includes the complete tree, to avoid cluttering these figures (already very compressed). Moreover, the current version of Figures 6 and 7 only include migration events with high support, and should therefore be more informative than the trees.

16. ✓ Figure 6+7: I suggest a removal of all non-significant migration arrows from the main figures as they can be confusing and lead to false interpretations.

Thank you for this visualization suggestion, we have removed all non-significant migration arrows from Figure 6 and Figure 7.

17. ✓ There is a typo in the legend of Figure 7, 0.PRE should be 1.PRE.

Thank you for catching this error. We have updated the legend of Figure 7 to indicate 1.PRE instead of 0.PRE.

18. ✓ There is a typo in the Conclusions section "Plague of Justinian (531CE)" should be 541.

Thank you for catching this error. We have updated the Conclusions to have the correct date (541 CE).

References

- Cui, Y., Yu, C., Yan, Y., Li, D., Li, Y., Jombart, T., Weinert, L. A., Wang, Z., Guo, Z., Xu, L., Zhang, Y., Zheng, H., Qin, N., Xiao, X., Wu, M., Wang, X., Zhou, D., Qi, Z., Du, Z., ... Yang, R. (2013). Historical variations in mutation rate in an epidemic pathogen, *Yersinia Pestis*. *Proceedings of the National Academy of Sciences*, *110*(2), 577–582. <https://doi.org/10.1073/pnas.1205750110>
- Damgaard, P. de B., Marchi, N., Rasmussen, S., Peyrot, M., Renaud, G., Korneliussen, T., Moreno-Mayar, J. V., Pedersen, M. W., Goldberg, A., Usmanova, E., Baimukhanov, N., Loman, V., Hedeager, L., Pedersen, A. G., Nielsen, K., Afanasiev, G., Akmatov, K., Aldashev, A., Alpaslan, A., ... Willerslev, E. (2018). 137 ancient human genomes from across the Eurasian steppes. *Nature*, *557*(7705), 369–374. <https://doi.org/10.1038/s41586-018-0094-2>
- Möller, S., du Plessis, L., & Stadler, T. (2018). Impact of the tree prior on estimating clock rates during epidemic outbreaks. *Proceedings of the National Academy of Sciences*, *115*(16), 4200–4205. <https://doi.org/10.1073/pnas.1713314115>
- Namouchi, A., Guellil, M., Kersten, O., Hänsch, S., Ottoni, C., Schmid, B. V., Pacciani, E., Quaglia, L., Vermunt, M., Bauer, E. L., Derrick, M., Jensen, A. Ø., Kacki, S., Cohn, S. K., Stenseth, N. C., & Bramanti, B. (2018). Integrative approach using *Yersinia pestis* genomes to revisit the historical landscape of plague during the Medieval Period. *Proceedings of the National Academy of Sciences*, *115*(50), E11790–E11797. <https://doi.org/10.1073/pnas.1812865115>
- Rascovan, N., Sjögren, K.-G., Kristiansen, K., Nielsen, R., Willerslev, E., Desnues, C., & Rasmussen, S. (2019). Emergence and Spread of Basal Lineages of *Yersinia pestis* during the Neolithic Decline. *Cell*, *176*(1–2), 295–305. <https://doi.org/10.1016/j.cell.2018.11.005>
- Rasmussen, S., Allentoft, M. E., Nielsen, K., Orlando, L., Sikora, M., Sjögren, K.-G., Pedersen, A. G., Schubert, M., Van Dam, A., Kapel, C. M. O., Nielsen, H. B., Brunak, S., Avetisyan, P., Epimakhov, A., Khalyapin, M. V., Gnuni, A., Kriiska, A., Lasak, I., Metspalu, M., ... Willerslev, E. (2015). Early divergent strains of *Yersinia Pestis* in eurasia 5,000 years ago. *Cell*, *163*(3), 571–582. <https://doi.org/10.1016/j.cell.2015.10.009>

- Ritchie, A. M., & Ho, S. Y. W. (2019). Influence of the tree prior and sampling scale on Bayesian phylogenetic estimates of the origin times of language families. *Journal of Language Evolution*, 4(2), 108–123. <https://doi.org/10.1093/jole/lzz005>
- Spyrou, M. A., Keller, M., Tukhbatova, R. I., Scheib, C. L., Nelson, E. A., Andrades Valtueña, A., Neumann, G. U., Walker, D., Alterauge, A., Carty, N., Cessford, C., Fetz, H., Gourvenec, M., Hartle, R., Henderson, M., von Heyking, K., Inskip, S. A., Kacki, S., Key, F. M., ... Krause, J. (2019). Phylogeography of the second plague pandemic revealed through analysis of historical *Yersinia Pestis* genomes. *Nature Communications*, 10(1), 4470. <https://doi.org/10.1038/s41467-019-12154-0>
- Spyrou, M. A., Tukhbatova, R. I., Wang, C.-C., Valtueña, A. A., Lankapalli, A. K., Kondrashin, V. V., Tsybin, V. A., Khokhlov, A., Kühnert, D., Herbig, A., Bos, K. I., & Krause, J. (2018). Analysis of 3800-Year-Old *Yersinia Pestis* genomes suggests bronze Age Origin for bubonic plague. *Nature Communications*, 9(1), 2234. <https://doi.org/10.1038/s41467-018-04550-9>
- Susat, J., Lübke, H., Immel, A., Brinker, U., Macāne, A., Meadows, J., Steer, B., Tholey, A., Zagorska, I., Gerhards, G., Schmölcke, U., Kalniņš, M., Franke, A., Pētersone-Gordina, E., Teßman, B., Tõrv, M., Schreiber, S., Andree, C., Bērziņš, V., ... Krause-Kyora, B. (2021). A 5,000-year-old hunter-gatherer already plagued by *Yersinia pestis*. *Cell Reports*, 35(13), 109278. <https://doi.org/10.1016/j.celrep.2021.109278>

Reviewers' comments:

Eaton and colleagues have included appreciable clarifications and analyses to their revision of the study "Plagued by a cryptic clock: Insight and issues from the global phylogeny of *Yersinia pestis*". I particularly appreciate the detailed description of the 601 genomes included in their *Y. pestis* dataset.

Yet, I find that four major points are still insufficiently addressed and some discussion statements misrepresent cited work. These are:

1. With regard to the long terminal branch lengths in 2.MED and 0.PE genomes, it has not been sufficiently clarified why those genomes are still used in the Bayesian phylogenetic analysis. Given the slow molecular clock of *Y. pestis*, it seems rather unlikely that those genomes do not affect the temporal signal of 2.MED and the resulting divergence date estimates. From the root-to-tip regression analysis, it is rather clear that strains with long terminal branches appear as outliers to the regression. A root-to-tip regression and formal temporal signal analysis excluding outlier genomes should be performed to test this possibility.

2. Regarding the divergence estimated of 1.PRE, the authors mention that the most ancestral genome in this clade is the LAI009 genome. Yet, in Figure S7, the maximum-clade-credibility tree of 1.PRE shows the LAI009 genome to be in a derived position, which misrepresents its true topology. The limited diversity in this clade, as well as the small number of derived substitutions in several 1.PRE genomes can cause such topology issues in BEAST. As a result of the artificially-derived positioning of LAI009, an older date can be estimated as TMRCA for the entire clade. The authors should use monophyletic clade priors to fix these issues in their analysis. A divergence estimate that contrasts the true substitution tree topology cannot be presented as a major result of the paper, especially given the historical implications with regard to pandemic origins outlined in the Discussion.

3. With regard to LAI009, the authors mention in their response that "Hypothetically, if the collection date was truly on the lower bound (1300 CE), it is possible that the MRCA could be 9.5 years before that (1290 CE). In this rough thought experiment, 1290 CE is contained in the confidence interval of 1214-1315 CE and is plausible (i.e. the branch lengths in the maximum-likelihood tree are congruent with the Bayesian estimates)". The authors entertain the possibility of a 1300 collection date but, in this assumption, fail to also acknowledge that LAI009 is only one SNP away from indistinguishable clone genomes associated with the Black Death of 1346-1353. In this regard, the authors statement seems to imply that the *Y. pestis* molecular clock in this part of the 1.PRE tree is consistent with one substitution every 46-53 years, which vastly contrasts their reported estimate of one substitution every 9.5 years. Given these priors, is a 1300 date for LAI009 realistic? Such an assumption would contrast some of the main findings of the study and, therefore, should be approached more holistically.

4. With regard to the discussion on pandemic origins, Eaton et al estimate an age range for the 1.PRE TMRCA, between 1214-1315, which does not overlap with the MRCA of Branches 1-4 published recently, that is ancestral to 1.PRE and is directly dated to the years 1338-1339. The authors should describe how are their results reconciled in light of recent data. Moreover, the new version of the Eaton et al study cites the paper describing the 1338-1339 *Y. pestis* genomes from Issyk Kul, however, in a misrepresentative manner as: (1) the genomes are described as falling close to the node of the Branch 1-4 polytomy, whereas the original cited paper mentions that they fall exactly on the node, and (2) the original cited paper does not represent 1338-1339 epidemic as the ultimate emergence event or geographic source, but rather points to the Tian Shan mountains during the beginning of the 14th century as the most likely initiation region and timing. Such misleading statements should be corrected by Eaton and colleagues in their Discussion. Moreover, the findings from 1338-1339 Issyk Kul paper

appear consistent with the migration events estimated in this study, where 1.PRE diverged from an ancestral 0.ANT population with "location being near Xinjiang (confidence: 0.64) which includes the Tien Shan mountains (Supplementary Figure S3)". In that regard, the authors should give credit to all previous studies that used similar datasets to propose this region as the Second Pandemic source.

We thank this reviewer for their repeatedly erudite and specific comments, which have made our manuscript a better paper. We address the last of the comments below (reviewers comments italicized, responses in blue).

Eaton and colleagues have included appreciable clarifications and analyses to their revision of the study "Plagued by a cryptic clock: Insight and issues from the global phylogeny of Yersinia pestis". I particularly appreciate the detailed description of the 601 genomes included in their Y. pestis dataset.

We thank the reviewer for this comment!

Yet, I find that four major points are still insufficiently addressed and some discussion statements is represent cited work. These are:

1. With regard to the long terminal branch lengths in 2.MED and 0.PE genomes, it has not been sufficiently clarified why those genomes are still used in the Bayesian phylogenetic analysis. Given the slow molecular clock of Y. pestis, it seems rather unlikely that those genomes do not affect the temporal signal of 2.MED and the resulting divergence date estimates. From the root-to-tip regression analysis, it is rather clear that strains with long terminal branches appear as outliers to the regression. A root-to-tip regression and formal temporal signal analysis excluding outlier genomes should be performed to test this possibility.

We have conducted a series of outlier analyses for 2.MED and still did not find temporal signal in this data set. Concretely, we removed the top 10% longest terminal branches and the root-to-tip regression still displays a negative slope and low R^2 value (Fig R1).

Figure R1. The left panel shows a screenshot from TempEst where the complete data for 2.MED is used. The right panel is for the same data set but with the 10% longest terminal branches have been removed.

We conducted the same series of analyses for 0.PE to remove the 10% longest terminal branches. In this case, removing such branches resulted in a weaker temporal signal, with a decrease in the R^2 and a negative regression slope (Fig R2). We explained our outlier analyses in the manuscript in **Estimating Rates of Evolutionary Change**.

Figure R2. The left panel shows a screenshot from TempEst where the complete data for 0.PE is used. The right panel is for the same data set but with the 10% longest terminal branches have been removed.

2. Regarding the divergence estimated of 1.PRE, the authors mention that the most ancestral genome in this clade is the LAI009 genome. Yet, in Figure S7, the maximum-clade-credibility tree of 1.PRE shows the LAI009 genome to be in a derived position, which misrepresents its true topology. The limited diversity in this clade, as well as the small number of derived substitutions in several 1.PRE genomes can cause such topology issues in BEAST. As a result of the artificially-derived positioning of LAI009, an older date can be estimated as TMRCA for the entire clade. The authors should use monophyletic clade priors to fix these issues in their analysis. A divergence estimate that contrasts the true substitution tree topology cannot be presented as a major result of the paper, especially given the historical implications with regard to pandemic origins outlined in the Discussion.

We conducted additional analyses to address this point. First, we estimated a tree using maximum likelihood and rooted it using LAI009. Second, we analysed the data using a constrained topology in BEAST, using the same model as the unconstrained analyses included in the manuscript. The maximum-clade credibility tree shows a similar branching pattern, albeit the uncertainty in node ages is wider (Figs R3 and R4). Specifically, the age of the root node is somewhat more uncertain than in the unconstrained model, but there is overlap in the posterior densities. Thus, the age of the root is robust to the exact position to whether the topology and outgroup are constrained or not. We have explained this point in the **Informative Rates and Dates** subsection.

Figure R3. Maximum-clade credibility tree for 1.PRE constrained to the topology inferred using maximum likelihood and with sample LAI009 (here labelled SAMEA5818806) as the outgroup. Blue bars correspond to error bars and can sit outside of nodes because the node heights are those from the tree (i.e. they have not been distorted to the posterior mean).

Figure R4. Posterior densities for the root height (i.e. years before the youngest sample) of 1.PRE with a fixed tree topology and LAI009 as the outgroup in green and under an unconstrained model in blue.

3. With regard to LAI009, the authors mention in their response that “Hypothetically, if the collection date was truly on the lower bound (1300 CE), it is possible that the MRCA could be 9.5 years before that (1290 CE). In this rough thought experiment, 1290 CE is contained in the confidence interval of 1214-1315 CE and is plausible (i.e. the branch lengths in the maximum-likelihood tree are congruent with the

Bayesian estimates)”. The authors entertain the possibility of a 1300 collection date but, in this assumption, fail to also acknowledge that LAI009 is only one SNP away from indistinguishable clone genomes associated with the Black Death of 1346-1353. In this regard, the authors statement seems to imply that the *Y. pestis* molecular clock in this part of the 1.PRE tree is consistent with one substitution every 46-53 years, which vastly contrasts their reported estimate of one substitution every 9.5 years. Given these priors, is a 1300 date for LAI009 realistic? Such an assumption would contrast some of the main findings of the study and, therefore, should be approached more holistically.

We agree with the reviewer on this point. We apologize for not making this clearer. We addressed this question, as a response to the reviewers’ previous comments- but did not include it verbatim in the new version of the manuscript for this exact reason. It’s a thought experiment but of course clearly there are issues at play here namely i) possibly incomplete genomes – although the coverage of LAI009 is excellent and more likely ii) the variable μ rate, of which this paper talks about ad nauseum. There are now several basal genomes that are a single or a few SNPs away from the true root or appear to sit on it (speaks to point 4 raised by the reviewer – see below) and thus equally to or at least all equally distant to the Black Death clones circulating in those first 5 years of the epidemic.

The section that pertains to dating the MRCA of 1.PRE in our manuscript reads as follows with suggested new sentence in red:

*Our estimates of the tMRCA for the First and Second Pandemics share a common theme, in that the genetic origins potentially pre-date the appearance of plague in traditional (i.e. European) historical narratives. For example, the earliest textual evidence of the Second Pandemic (1.PRE) in Europe comes from the Black Death (1346)². However, we estimate the mean tMRCA of this population to be earlier, between 1214 and 1315 CE. Similarly, the first recorded outbreaks of plague during the First Pandemic (O.ANT4) come from the Plague of Justinian (541 CE)⁴². Instead, we estimate that the strains of *Y. pestis* associated with this pandemic shared a common ancestor between 272 and 465 CE.*

One explanation for these disparate timelines is uncertainty in estimating the root position when only samples from within a population are used. This pertains specifically to the Second Pandemic (1.PRE), where basal sequences have near-zero branch lengths in the maximum-likelihood phylogeny (Figure 1A) which leads to poor branch support in the maximum-clade-credibility tree (Supplementary Figure S7). This discrepancy in dates could suggest that our estimates are artifactually old, as the most basal strain (at the time of this work LAI009) is placed in a more derived position (farther from the root) in the maximum-clade-credibility tree, which is surprising given that it is only a single SNP different (at the core genome level) from the Black Death clonal strains that emerge in 1347/8. To resolve this, further testing is required to assess whether outgroup populations (ex. O.ANT4) can be included to improve the tMRCA estimation without destabilizing the molecular clock model.

*A second explanation for our earlier dates is tip date uncertainty. The radiocarbon estimates for the majority of ancient *Y. pestis* samples have confidence intervals of ± 50 years or more. As we only used the mean sampling age for molecular clock models, it’s possible that the true tMRCA intervals are larger and do overlap with historical estimates. How much uncertainty can be included in molecular clock models for *Y. pestis*, while still achieving convergence of parameter estimates, remains to be tested.*

A third explanation is geographic sampling bias, as western European sources dominate both the genetic and historical record. Recent historical scholarship has contested Eurocentric timelines^{28,43} by demonstrating the presence of plague in western Asia far earlier than previously thought. Arabic

historical chronicles suggest that the Second Pandemic may have begun as early as the 13th century⁴⁴. Genetic dating appears to support these historical critiques, by expanding the timelines of past pandemics to make space for more diverse historical narratives.

4. With regard to the discussion on pandemic origins, Eaton et al estimate an age range for the 1.PRE tMRCA, between 1214-1315, which does not overlap with the MRCA of Branches 1-4 published recently, that is ancestral to 1.PRE and is directly dated to the years 1338-1339. The authors should describe how are their results reconciled in light of recent data. Moreover, the new version of the Eaton et al study cites the paper describing the 1338-1339 *Y. pestis* genomes from Issyk Kul, however, in a misrepresentative manner as: (1) the genomes are described as falling close to the node of the Branch 1-4 polytomy, whereas the original cited paper mentions that they fall exactly on the node, and (2) the original cited paper does not represent 1338-1339 epidemic as the ultimate emergence event or geographic source, but rather points to the Tian Shan mountains during the beginning of the 14th century as the most likely initiation region and timing. Such misleading statements should be corrected by Eaton and colleagues in their Discussion. Moreover, the findings from 1338-1339 Issyk Kul paper appear consistent with the migration events estimated in this study, where 1.PRE diverged from an ancestral 0.ANT population with “location being near Xinjiang (confidence: 0.64) which includes the Tien Shan mountains (Supplementary Figure S3)”. In that regard, the authors should give credit to all previous studies that used similar datasets to propose this region as the Second Pandemic source.

We thank the author for their salient points here as well. Yes, the original citation does speak to the reconstructed sequences falling on the node not derived. I do think given the coverage of these genomes (I've learned this painfully in our first Justinian genome) that it may be prudent to wait for deeper coverage (which I presume/hope will come from these authors from additional samples). Some people have suggested that one or two SNPs (in the supplementary files) which have been not called in the consensus of these genomes, could (obviously) move the sequences off the node and thus there is some debate surrounding their exact position and hence the debate on the timing remains perhaps slightly tenuous. We haven't investigated those here in addition to this work so can't speak to it personally. As such we agree we have now more appropriately framed the work in the following way but reworking the paragraph on these remains as such:

Recently published aDNA data⁴⁵ characterizes *Y.pestis* genomes from two individuals from cemeteries located near Lake Issyk-Kul, Kyrgyzstan. These genomes fall on the node of the polytomy and are putatively dated to 1338-1339 and help constrain the clock by yielding an MRCA for branches 1-4 to between 1316-1345, which postdates our estimated age range of 1214-1315 for just the 1.PRE – second pandemic clade. Clearly, at this proximity to the true node, the variance on the clock has a huge effect on estimation of a MRCA. The only way to properly calibrate it is with well dated sequences as are those recently published from skeletal remains from Lake Issyk-Kul⁴⁵.

And later in the ms:

For example, we estimated the tMRCA of the Second Pandemic to be between 1214 and 1315 CE which pre-dates the Black Death (1346 - 1353 CE) and pre-dates a recent estimate of the MRCA for the polytomy from *Yersinia pestis* genomes reconstructed from Issyk-Kul, Kyrgyzstan⁴⁵. Refinement of these

early events and nodal MRCA's will require more ancient DNA data as the clock is too variable for this resolution

As regards the origins narrative. The reviewer is correct. We have rephrased the following sentences to read as follows:

The ancestral province was poorly resolved, with the most likely location being near Xinjiang (confidence: 0.64) which includes the Tien Shan mountains (Supplementary Figure S3) which is in congruence with recently published data⁴⁵.